# DAF: DYNAMIC ADAPTIVE FINE-TUNING OF VISION TRANSFORMERS

## ABSTRACT

Parameter-Efficient Fine-Tuning (PEFT) is essential for training large Vision Transformers (ViTs), yet existing methods are fundamentally constrained by a static allocation paradigm, where trainable parameters are fixed before training. We argue this static approach overlooks the evolving optimization priorities of a model during learning, thereby limiting its final performance under a constrained parameter budget. Inspired by the sparse dynamic activation mechanism of neurons in the brain, we introduce a novel dynamic reconfiguration paradigm for PEFT and propose a framework named Dynamic Adaptive Fine-tuning (DAF). The core of DAF lies in its ability to periodically evaluate, select, and reshape its trainable structure during training. It employs our proposed context-aware decoupled sensitivity analysis method to purely assess the backbone network's potential while preserving the full learning context. Subsequently, it executes the proposed Rebuild-and-Refocus update strategy. This strategy uniquely preserves learned knowledge by freezing outdated fine-tuning modules while decisively reallocating the entire parameter budget to newly identified critical regions. Extensive experiments on several highly challenging vision benchmarks show that the DAF framework not only significantly outperforms mainstream static PEFT methods but also achieves SOTA performance. Our work fundamentally challenges the static nature of the PEFT field and opens a new avenue for adapting large pretrained models more intelligently and efficiently. The code is available at `https://anonymous.4open.science/r/DAF-9372`.

## 1 INTRODUCTION

Large-scale pretrained vision models, particularly ViT (Dosovitskiy et al., 2021), have demonstrated remarkable generalization capabilities in many downstream visual tasks. The standard paradigm for adapting these powerful models to specific tasks is full fine-tuning. However, this approach requires storing a complete copy of the model for each task, and with the dramatic growth in the scale of the model (Zhai et al., 2022), the associated high storage and computational costs have become prohibitive. PEFT has emerged (Hu et al., 2022; Jia et al., 2022) to address this challenge, which tunes only a small fraction of the model's parameters, achieving performance comparable or even superior to that of full fine-tuning while significantly reducing resource consumption.

Existing PEFT methods largely follow a static allocation paradigm. One class of methods, such as Adapter (Houlsby et al., 2019), LoRA (Hu et al., 2022), and Visual Prompt Tuning (VPT) (Jia et al., 2022), typically relies on human prior knowledge to insert trainable modules at task-agnostic, fixed locations. Another class of methods attempts to adaptively select fine-tuning parameters for specific tasks (He et al., 2023). However, whether based on heuristic rules or a one-shot sensitivity analysis, these methods share a fundamental limitation: the locations and structures of all trainable parameters are determined once before training and remain unchanged throughout the entire fine-tuning process. Recently, although methods like VQT (Tu et al., 2023) and SynQT (Zhang et al., 2024a) have made new progress in how to utilize intermediate representations, the tuning structures they introduce are also fixed after training begins. This static assumption overlooks a critical fact: as the model progressively learns and adapts to the downstream task, its internal knowledge bottlenecks and optimization priorities dynamically evolve. A module that is crucial in the early stages of training may no longer be key to performance improvement later on; conversely, new bottlenecks

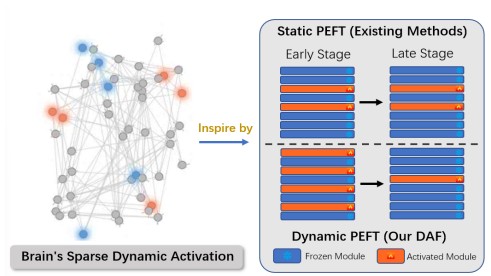

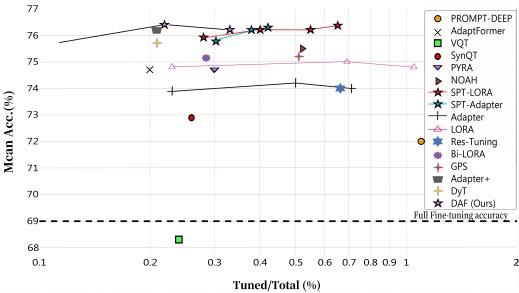

Figure 1: Inspired by the brain's sparse and dynamic activation(Chen et al., 2024), the DAF framework periodically reconfigures its trainable structure. This contrasts with static PEFT methods that use fixed modules.

Figure 2: Performance comparison on the VTAB-1k benchmark. The DAF achieves SOTA performance against various static PEFT methods with remarkable parameter efficiency (tuning only 0.22% of parameters).

will emerge. Therefore, any static allocation strategy cannot optimally utilize the limited parameter budget to adapt to the dynamic changes in the model's own learning state.

This fundamental limitation of the static assumption becomes particularly prominent when contrasted with the operational mechanism of nature's most efficient learning system—the biological brain. Learning in the nervous system is not a fixed, predetermined process. Instead, it is a highly dynamic process of remodeling. When faced with new knowledge or tasks, the brain does not uniformly activate all neurons. Rather, it employs a mechanism of sparse activation to selectively engage the specific neural circuits most relevant to the current stimulus (Poo et al., 2023). More importantly, through synaptic plasticity, the connection strengths between neurons are dynamically strengthened or weakened based on experience. This continuous structural adaptation is the key to achieving efficient, lifelong learning in biological intelligence.

Inspired by this biological mechanism (Payeur et al., 2023), we argue that a more ideal fine-tuning framework should be able to dynamically perceive and adapt to the model's evolution, as shown in Figure 1. To this end, we are the first to propose a novel dynamic reconfiguration paradigm for PEFT and design a framework named DAF to implement it. The core of DAF is a periodic perceive-decide-execute cycle. In each dynamic cycle, DAF initiates a three-stage process. First, it perceives the model's state by employing our proposed context-aware decoupled sensitivity analysis method, which accurately evaluates the potential of the underlying backbone within the context of all previously learned knowledge. Subsequently, it decides on the most critical components for the current learning stage using a focused Top-K elite selection mechanism. Finally, DAF executes a decisive reconfiguration by adopting our proposed Rebuild-and-Refocus strategy. This strategy thoroughly reorganizes the model's fine-tuning structure by freezing outdated modules while activating new ones, thereby concentrating all training resources on newly identified critical regions. This mechanism enables the model to shed the burden of training less relevant modules and adapt to new learning bottlenecks as quickly as possible. We conduct extensive experiments on multiple challenging public benchmarks. As illustrated in Figure 2, the compelling experimental results demonstrate that the DAF framework not only significantly outperforms mainstream static PEFT methods but also achieves SOTA level performance, validating the superiority of the dynamic paradigm. The main contributions consist of the following three aspects:

- We propose the first Dynamic Reconfiguration paradigm for PEFT, which fundamentally challenges the static nature of the immutable fine-tuning structures in existing methods.

- We design and implement a complete framework named DAF, the core of which is a sophisticated Rebuild-and-Refocus strategy. This strategy uniquely preserves learned knowledge in previously important modules by freezing them, while decisively reallocating the training budget to new bottlenecks.

- We design a context-aware decoupled sensitivity analysis method to solve the signal noise problem in dynamic decision-making. This method temporarily freezes existing fine-tuning modules on the complete model, enabling a pure assessment of the backbone network's potential while preserving the full learning context.

## 2 RELATED WORK

**Parameter-Efficient Fine-Tuning.** Existing PEFT methods can be broadly categorized into three types based on how they introduce trainable parameters. Addition-based Tuning adapts pretrained models by injecting new modules or prompts. Among these, Adapter (Houlsby et al., 2019) serially inserts small bottleneck layers, whereas AdaptFormer (Chen et al., 2022) places them in parallel with the FFN. This line of work has seen continued refinement, with Adapter+ (Steitz & Roth, 2024) optimizing the static adapter configuration, and methods like Mona (Yin et al., 2024) and LoRand (Yin et al., 2023b) proposing new adapter architectures for complex dense prediction tasks. Another popular branch is VPT (Jia et al., 2022), which adds learnable prompt tokens. Specification-based Tuning selectively fine-tunes a small subset of the model's intrinsic parameters. For example, BitFit (Zaken et al., 2022a) tunes only the bias terms, and SSF (Lian et al., 2022) learns to scale and shift parameters. Reparameterization-based tuning methods, notably LoRA (Hu et al., 2022), approximate weight updates using trainable low-rank matrices, which can be merged at inference. This approach has also been recently enhanced by methods like DoRA (Liu et al., 2024a), which decomposes weights into magnitude and direction. Despite the success of these methods, their decisions on 'where to fine-tune' largely rely on task-agnostic heuristics. To address this, some works explore adaptive parameter selection. For instance, GPS (Zhang et al., 2024b) and SPT (He et al., 2023) proposed identifying the most important parameters for a task via a one-time sensitivity analysis before fine-tuning begins. However, a common thread among all these methods (including heuristic-based, architecture-based, and selection-based) is their adherence to a static allocation paradigm. Our work fundamentally challenges this static assumption, arguing that the fine-tuning structure itself should evolve with the training process.

**Dynamic Model Adaptation.** The concept of dynamics has been explored in other areas, but the objectives differ fundamentally from our work. One line of research focuses on Inference-Stage Dynamics to improve computational efficiency. For instance, DynamicViT (Rao et al., 2021) and DVT (Wang et al., 2021) dynamically prune tokens. More recently, Sparse-Tuning (Liu et al., 2024b) and DyT (Zhao et al., 2024) combine sparsification with PEFT to optimize inference. The core objective of these methods is to accelerate inference, whereas DAF focuses on making the trainable structure dynamic during the training stage to improve final model performance. Another line of work applies dynamic ideas to *Continual Learning* to mitigate catastrophic forgetting. For example, some methods dynamically allocate new parameters for each new task (Wang et al., 2024). Recently, SD-LoRA (Wu et al., 2024) explored decoupling magnitude and direction for class-incremental learning. The goal of these works is to balance stability and plasticity when learning a sequence of discrete tasks, whereas DAF focuses on dynamic adaptation within a single task. A third category focuses on static adaptation while improving training efficiency. Head2Toe (Evci et al., 2022) and LST (Sung et al., 2022) train lightweight side-networks. Similarly, E³VA (Yin et al., 2023a) proposes a parallel adapter highway to reduce training time and memory, but its focus is on computational efficiency rather than adaptive learning. VQT (Tu et al., 2023) and SynQT (Zhang et al., 2024a) introduce learnable queries. While effective, their interaction mechanisms remain fixed. Applying the concept of dynamics to the fine-tuning process of a single downstream task itself remains a largely unexplored direction. A notable exception is AdaLoRA (Zhang et al., 2023), which adaptively prunes the rank of LoRA modules during training based on an importance score. However, this method focuses on pruning (reducing) a budget from a large initial rank. In contrast, DAF introduces a new paradigm of dynamic reconfiguration: it periodically and adaptively re-allocates its entire fixed-size PEFT structure by freezing outdated modules and activating new ones. The DAF framework aims to fill this critical gap, positing that this intra-task training dynamism is key to achieving a deeper and more efficient model adaptation.

## 3 METHOD

In the section, we introduce the preliminaries of ViT and LoRA, and then design on the overall structure of the DAF framework, including its core techniques: Context-Aware Decoupled Sensitivity Analysis and the Dynamic Reconfiguration mechanism.

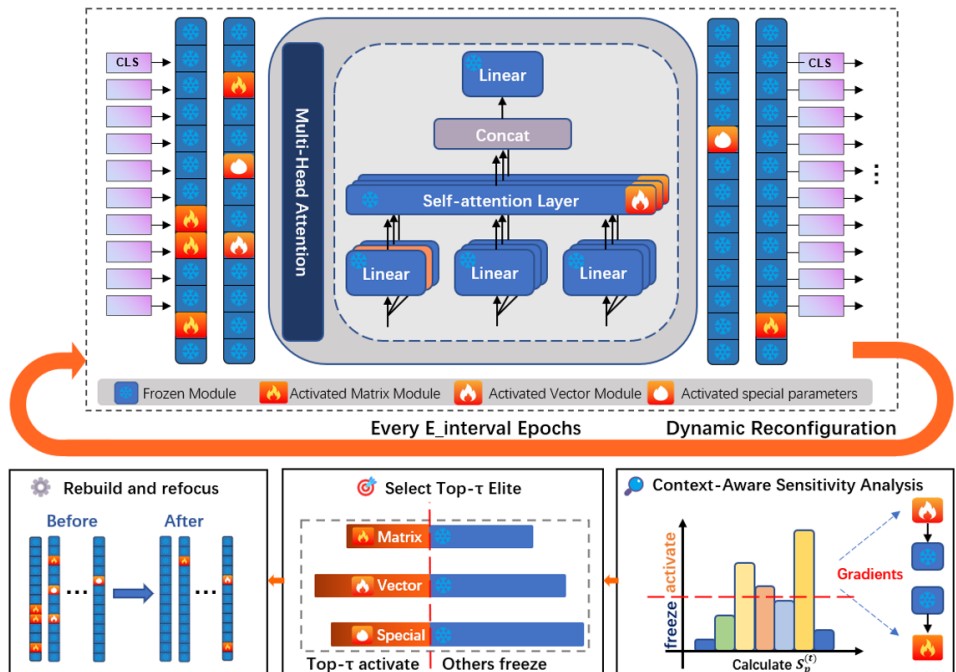

Figure 3: Overall framework of DAF. DAF periodically reconfigures the trainable structure of a pre-trained ViT during fine-tuning. At each dynamic analysis point, it executes a three-stage cycle: (1) Perceive the model state via Context-Aware Decoupled Sensitivity Analysis, (2) Decide on the most critical parameters (Matrix, Vector, and Special) using a budget-based elite selection, (3) Execute a Rebuild-and-Refocus strategy to update the set of active modules for the next training interval.

### 3.1 PRELIMINARIES

**Vision Transformer.** A standard ViT model consists of a patch embedding layer and $L$ stacked Transformer Blocks. Each Transformer Block $l \in \{1, \ldots, L\}$ typically includes a Multi-Head Self-Attention (MSA) module and a Feed-Forward Network (FFN). During fine-tuning, the vast majority of the ViT's parameters, denoted as $\theta_{\text{vit}}$, remain frozen. LoRA is an efficient structured fine-tuning technique. For a pretrained weight matrix $W_0 \in \mathbb{R}^{d \times k}$, LoRA approximates its update, $\Delta W$, by introducing two trainable low-rank matrices, $A \in \mathbb{R}^{d \times r}$ and $B \in \mathbb{R}^{r \times k}$ (where the rank $r \ll \min(d, k)$). During the forward pass, the output $y$ of the layer is computed as follows:

$$y = W_0 x + \Delta W x = W_0 x + BAx \tag{1}$$

where only $A$ and $B$ are trainable.

### 3.2 THE DAF FRAMEWORK: A DYNAMIC RECONFIGURATION PARADIGM

The overall framework of DAF follows a periodic cycle of perceive-decide-execute, which we term Dynamic Reconfiguration. Unlike static methods that determine all trainable parameters at once before training, DAF repeatedly executes this cycle throughout the training process to achieve continuous optimization of the model's fine-tuning structure. We illustrate the overall framework of DAF in Figure 3. As shown in the figure, the main training flow involves a pre-trained ViT where individual layers can be dynamically frozen or have their internal parameters activated. This activation state is not fixed at every dynamic analysis point (e.g., every $E_{\text{interval}}$ epoch) and the Dynamic Reconfiguration module is triggered. This module executes a three-stage process: it first perceives the model's state through dynamic sensitivity analysis, then decides on the new set of elite parameters to train, and finally executes the structural update via the Rebuild-and-Refocus strategy, which ensures that the limited training budget is allocated to the most critical parts.

Assuming the total number of training epochs is $E_{\text{total}}$ and the dynamic analysis interval is $E_{\text{interval}}$, DAF performs the following core operations at each dynamic analysis point $t \in \{E_{\text{interval}}, 2E_{\text{interval}}, \ldots\}$:

- **Perceive:** Accurately assess the potential of the underlying backbone network within the complete learning context of the current model (see Section 3.3).

- **Decide:** Based on the assessment, employ a budget-based elite selection mechanism to identify the most important set of parameters $\mathcal{P}_t^*$ for the current stage (see Section 3.4).

- **Execute:** Adopt a Rebuild-and-Refocus strategy to reconstruct the model's structure, ensuring it only contains the tuning modules corresponding to the set $\mathcal{P}_t^*$, and perform weight migration (see Section 3.4).

## 3.3 CONTEXT-AWARE DECOUPLED SENSITIVITY ANALYSIS

Providing a precise navigation signal for dynamic reconfiguration is key to DAF's success. We found that directly analyzing a model that includes active LoRA modules introduces signal noise, as the training state such as high gradients of the LoRA modules themselves can dominate the analysis. To address this, we propose a context-aware decoupled dynamic analysis method. At each dynamic analysis point $t$, we define the main model being trained as $\mathcal{M}_t$, with parameters comprising the backbone weights $\theta_{\text{bb}}^{(t)}$ and all existing LoRA module parameters $\theta_{\text{lora}}^{(t)}$. The analysis begins by temporarily freezing LoRA modules, we iterate through $\mathcal{M}_t$ and set the parameters of all existing LoRA modules to a non-trainable state. Next, we perform an end-to-end gradient computation on the complete model. Since the LoRA modules still participate in the forward pass, they provide the correct, evolving learning context for the backbone's gradient calculation. During the backward pass, the gradient flow is influenced by all LoRA modules but ultimately accumulates only on the backbone parameters, yielding a pure gradient information. For any backbone parameter $w_p \in \theta_{\text{bb}}^{(t)}$, its sensitivity $s_p^{(t)}$ can be approximated by drawing inspiration from the model pruning method:

$$s_p^{(t)} = \left| \frac{\partial \mathcal{L}(\mathcal{D}; \mathcal{M}_t(\theta_{\text{bb}}^{(t)}, \theta_{\text{lora}}^{(t)}))}{\partial w_p} \cdot w_p \right| \tag{2}$$

where $\mathcal{L}$ is the task loss function and $\mathcal{D}$ is a small batch of data used for analysis. Because the gradients of LoRA parameters are disabled, this score reflects the purest potential signal of the backbone network. This dynamic, context-aware approach fundamentally differs from static methods that perform a one-shot analysis of the original unmodified backbone before any training begins.

## 3.4 DYNAMIC RECONFIGURATION: BUDGET-BASED ELITE SELECTION AND REBUILD-AND-REFOCUS UPDATE

After obtaining the sensitivity scores for all backbone parameters, DAF employs a refined, budget-based elite selection mechanism instead of a simple global Top-K selection.

**Parameter Categorization.** To prevent the selection from being dominated by superior matrix parameters and ensure a functionally diverse set of tunable parameters is chosen, we first classify all backbone parameters to be analyzed into three categories: Matrix Parameters ($\mathcal{P}_{\text{mat}}$), which are the primary candidates for LoRA tuning. Vector Parameters ($\mathcal{P}_{\text{vec}}$), such as LayerNorm weights and biases, and Special Parameters ($\mathcal{P}_{\text{spec}}$), like cls_token and pos_embed.

**Budget-Based Elite Selection.** We first define an overall parameter budget, denoted by a ratio $\tau$ (e.g., $\tau = 0.2$), which represents the target fraction of total possible PEFT parameters to be activated. To determine the specific parameter count budget for each category, this overall ratio $\tau$ is combined with fixed allocation ratios for Matrix Parameters ($\mathcal{P}_{\text{mat}}$), Vector Parameters ($\mathcal{P}_{\text{vec}}$) and Special Parameters ($\mathcal{P}_{\text{spec}}$). This partitioned budget strategy is crucial as it ensures a balanced selection of diverse parameter types, preventing the sensitivity scores of large-scale matrix parameters from dominating and overshadowing smaller, yet functionally critical, vector or special parameters. Within each category's specifically allocated budget, we then rank parameters by their sensitivity scores $s_p^{(t)}$ and select the top-ranking ones to form the active sets $\mathcal{A}_{\text{mat}}^{(t)}$, $\mathcal{A}_{\text{vec}}^{(t)}$, and $\mathcal{A}_{\text{spec}}^{(t)}$. The final set of parameters to be fine-tuned in the current cycle is the union $\mathcal{P}_t^* = \mathcal{A}_{\text{mat}}^{(t)} \cup \mathcal{A}_{\text{vec}}^{(t)} \cup \mathcal{A}_{\text{spec}}^{(t)}$. We add a constraint that only parameters with non-zero gradients are considered.

**Rebuild-and-Refocus Update.** This mechanism ensures both learning continuity and adaptive resource allocation through a two-step process. First, for model reconstruction, we create a new, clean

model instance, $\mathcal{M}_{t+1}$. This new model instantiates LoRA structures for all modules $m$ in the union of the previous and current active sets, $\mathcal{A}_{\text{mat}}^{(t-1)} \cup \mathcal{A}_{\text{mat}}^{(t)}$. Second, we perform a meticulous weight migration to transfer knowledge from the old model $\mathcal{M}_t$. For any parameter $w'_p$ in the new model $\mathcal{M}_{t+1}$:

$$
w'_p \leftarrow \begin{cases} w_p & \text{if } p \text{ is a backbone parameter} \\ w_p & \text{if } p \text{ is in a LoRA module for } m \in \mathcal{A}_{\text{mat}}^{(t-1)} \\ \text{Re-initialized} & \text{if } p \text{ is in a LoRA module for } m \in \mathcal{A}_{\text{mat}}^{(t)} \setminus \mathcal{A}_{\text{mat}}^{(t-1)} \end{cases} \tag{3}
$$

where $w_p$ is the corresponding parameter from the old model $\mathcal{M}_t$. Crucially, after weight migration, the Refocus step redefines the entire set of trainable parameters for the next training interval. For the matrix parameters, only the LoRA modules corresponding to the new active set $\mathcal{A}_{\text{mat}}^{(t)}$ are enabled for training. Any LoRA module from the previous step that is no longer selected ($m \in \mathcal{A}_{\text{mat}}^{(t-1)} \setminus \mathcal{A}_{\text{mat}}^{(t)}$) is immediately frozen, preserving its acquired knowledge while freeing up resources. In parallel, this update logic extends to the intrinsic backbone parameters: those selected for the new elite vector and special sets, $\mathcal{A}_{\text{vec}}^{(t)}$ and $\mathcal{A}_{\text{spec}}^{(t)}$, are marked as trainable, while any previously trained vector or special parameters that are no longer part of the elite sets are frozen.

This comprehensive process ensures that knowledge from the backbone and all previously learned modules is inherited, while training resources are decisively refocused on the newly identified critical regions across all parameter types. Through this cycle, DAF ensures that its fine-tuning structure is tailored to the current learning state at each stage, thereby achieving maximal adaptability.

**Zero-Overhead Inference.** Upon the completion of training, DAF employs a re-parameterization technique. All learned LoRA parameters (matrices $A$ and $B$), regardless of the training stage in which they were activated, are mathematically merged into the backbone weights via $W_{\text{final}} = W_0 + BA$. Consequently, the final model is architecturally identical to the original ViT, requiring no extra storage for historical parameters and incurring zero additional computational cost during inference.

The complete process of the DAF algorithm is detailed in Algorithm 1.

---

**Algorithm 1** DAF: Dynamic Adaptive Fine-tuning

---

**Require:** Pretrained ViT model $\mathcal{M}_0$, Total epochs $E_{\text{total}}$, Dynamic interval $E_{\text{interval}}$, Budget $\tau$.
1: Initialize model $\mathcal{M} \leftarrow \mathcal{M}_0$.
2: Perform initial sensitivity analysis on $\mathcal{M}$ to get initial active sets $\mathcal{A}_{\text{mat}}^{(0)}, \mathcal{A}_{\text{vec}}^{(0)}, \mathcal{A}_{\text{spec}}^{(0)}$.
3: Rebuild model $\mathcal{M}$ with LoRA modules for $\mathcal{A}_{\text{mat}}^{(0)}$ and enable gradients for other active parameters.
4: Initialize Optimizer $\mathcal{O}$ for all trainable parameters in $\mathcal{M}$.
5: **for** epoch $t = 1$ to $E_{\text{total}}$ **do**
6:     Train model $\mathcal{M}$ for one epoch using Optimizer $\mathcal{O}$.
7:     **if** $t \ (\text{mod } E_{\text{interval}}) = 0$ **then**
8:         ▷ *Perceive: Context-Aware Decoupled Sensitivity Analysis*
9:         Temporarily freeze all existing LoRA modules in $\mathcal{M}$ (`requires_grad=False`).
10:        Compute backbone sensitivity scores $s_p^{(t)}$ on $\mathcal{M}$ using Eq. equation 2.
11:        Unfreeze LoRA modules.
12:        ▷ *Decide & Execute: Rebuild-and-Refocus*
13:        Save current active sets as $\mathcal{A}_{\text{mat}}^{(t-1)}, \mathcal{A}_{\text{vec}}^{(t-1)}, \mathcal{A}_{\text{spec}}^{(t-1)}$.
14:        Determine new active sets $\mathcal{A}_{\text{mat}}^{(t)}, \mathcal{A}_{\text{vec}}^{(t)}, \mathcal{A}_{\text{spec}}^{(t)}$ based on top $\tau$ sensitive parameters.
15:        Create new model $\mathcal{M}_{\text{new}}$ with LoRA modules for all $m \in \mathcal{A}_{\text{mat}}^{(t-1)} \cup \mathcal{A}_{\text{mat}}^{(t)}$.
16:        Perform weight migration from $\mathcal{M}$ to $\mathcal{M}_{\text{new}}$ using Eq. equation 3.
17:        Set only parameters corresponding to $\mathcal{A}_{\text{mat}}^{(t)}, \mathcal{A}_{\text{vec}}^{(t)}, \mathcal{A}_{\text{spec}}^{(t)}$ as trainable in $\mathcal{M}_{\text{new}}$.
18:        $\mathcal{M} \leftarrow \mathcal{M}_{\text{new}}$.
19:        Re-initialize Optimizer $\mathcal{O}$ for the new trainable parameters in $\mathcal{M}$.
20:     **end if**
21: **end for**
22: **return** Trained model $\mathcal{M}$.

---

## 4    EXPERIMENTS

In this section, we conduct extensive experiments to evaluate the proposed DAF framework. First, we compare DAF against the mainstream and latest PEFT methods across diverse benchmarks, then conduct comprehensive ablation studies on its core components, and finally analyze how the key hyperparameters influence its performance.

### 4.1    EXPERIMENTAL SETUP

**Datasets.** To ensure a comprehensive evaluation, we conduct experiments across a diverse range of visual tasks. For classification, we utilize the standard Fine-Grained Visual Classification (FGVC) benchmark and the large-scale Visual Task Adaptation Benchmark (VTAB-1k) (Zhai et al., 2019). To further demonstrate the versatility of DAF on complex dense prediction tasks, we extend our evaluation to Object Detection on MS COCO (Lin et al., 2014) and Semantic Segmentation on ADE20K (Zhou et al., 2017). Further details are provided in Appendix A.1.

**Implementation Details.** Our primary experiments utilize a ViT-B/16 backbone pre-trained on ImageNet-21k. To verify architectural generalization, we also employ Hierarchical Transformers (Swin-B/L) and CNN-based architectures (ConvNeXt-B). We use the AdamW optimizer on NVIDIA RTX 4090 GPUs. Detailed hyperparameters are listed in Appendix A.2.

**Baselines.** We compare DAF against a comprehensive suite of PEFT methods, ranging from classic approaches to the latest SOTA. These baselines cover three primary paradigms: (1) Addition-based methods, such as Adapter (Houlsby et al., 2019), AdaptFormer (Chen et al., 2022); (2) Reparameterization-based methods, such as LoRA (Hu et al., 2022); and (3) Prompt-based methods, such as VPT (Jia et al., 2022), NOAH (Zhang et al., 2022). Crucially, we include strong competitors from 2023-2024, such as SPT (He et al., 2023), VQT (Tu et al., 2023), Res-Tuning (Jiang et al., 2023), LoRand (Yin et al., 2023b) Bi-LoRA  (Jie et al., 2023), PYRA (Xiong et al., 2024), DyT (Zhao et al., 2024), Adapter+  (Steitz & Roth, 2024), GPS (Zhang et al., 2024b), SynQT (Zhang et al., 2024a), and Mona (Yin et al., 2024), to ensure a rigorous comparison against the current research frontier. A "Static DAF" baseline is also introduced to isolate the benefits of our dynamic mechanism.

### 4.2    MAIN RESULTS ON STANDARD BENCHMARKS

We first compare DAF with baseline methods under the standard setting using a supervised pre-trained ViT-B/16 backbone. The average accuracy on FGVC and VTAB-1k is presented in Table 1.

As delineated in Table 1, the proposed DAF framework achieves SOTA performance, outperforming all static PEFT baselines, including recent strong competitors like GPS (Zhang et al., 2024b) and SPT-LoRA (He et al., 2023) on both benchmarks. This highlights the significant advantage of dynamically reallocating parametric resources during training over a fixed, pre-determined fine-tuning strategy. Furthermore, the performance gap between DAF and our Static DAF baseline directly validates that the observed gains are attributable to the dynamic reconfiguration mechanism. To further substantiate the stability and convergence superiority of this dynamic process, we provide a visual analysis of training loss trajectories in Appendix A.3. For a more granular analysis, we present the detailed per-task results on both benchmarks in Appendix A.4 and provide the corresponding dynamic behavior visualizations for each task in Appendix A.5. In addition to model performance, we also evaluate the computational efficiency of our method. As detailed in Appendix A.6 and Appendix A.7, DAF demonstrates a highly competitive efficiency profile, achieving its superior performance with a minimal parameter budget and no additional inference overhead.

### 4.3    VERSATILITY IN SELF-SUPERVISED PRE-TRAINING PARADIGMS

To rigorously test the versatility and robustness of DAF, we evaluated DAF on ViT-B/16 models pre-trained with two distinct self-supervised paradigms: Masked Autoencoders (MAE) and Momentum Contrast v3 (MoCo v3). Theoretically, the specialized features learned through these paradigms can result in highly varied parameter sensitivities across different downstream tasks. This characteristic poses a significant challenge for static PEFT methods, as their fixed allocation of trainable parameters may struggle to adapt to such shifting optimization requirements.

Table 1: Overall performance comparison on FGVC and VTAB-1k benchmarks (ViT-B/16, ImageNet-21k pre-trained). Accuracy is Top-1 Avg. (%). 'Tuned/Total' denotes the fraction of trainable parameters. We highlight the **best** and the second-best results. Recent SOTA methods (2023-2024) are included as per reviewer feedback.

| | FGVC | | VTAB-1k | | | | |
|---|---|---|---|---|---|---|---|
| Method | Tuned/Total (%) | Mean Acc. (%) | Tuned/Total (%) | Natural | Specialized | Structured | Mean Acc. (%) |
| Full Fine-tuning | 100 | 88.5 | 100 | 75.9 | 83.4 | 47.6 | 69.0 |
| *Static PEFT Baselines* | | | | | | | |
| Adapter-8 (Houlsby et al., 2019) | 0.39 | 85.5 | 0.23 | 79.0 | 84.1 | 58.5 | 73.9 |
| Adapter-32 (Houlsby et al., 2019) | 0.95 | 85.6 | 0.71 | 79.6 | 84.0 | 58.3 | 74.0 |
| LoRA-8 (Hu et al., 2022) | 0.55 | 86.0 | 0.23 | 79.5 | 84.6 | 60.5 | 74.9 |
| LoRA-16 (Hu et al., 2022) | 0.90 | 84.8 | 0.69 | 79.8 | 84.9 | 60.2 | 75.0 |
| VPT-Deep (Jia et al., 2022) | 0.35 | 83.8 | 0.32 | 78.5 | 82.4 | 55.0 | 72.0 |
| AdaptFormer (Chen et al., 2022) | 0.23 | 86.1 | **0.20** | 80.5 | 84.9 | 58.8 | 74.7 |
| NOAH (Zhang et al., 2022) | 0.50 | 89.2 | 0.52 | 80.2 | 84.9 | 61.3 | 75.5 |
| *Recent Static SOTA Baselines* | | | | | | | |
| VQT (Tu et al., 2023) | 0.30 | 82.5 | 0.24 | 76.0 | 80.2 | 46.3 | 68.3 |
| SPT-Adapter (He et al., 2023) | 0.41 | 89.5 | 0.30 | 81.3 | 85.3 | 60.8 | 75.8 |
| SPT-LoRA (He et al., 2023) | 0.41 | 89.3 | 0.31 | 81.5 | 85.6 | 60.7 | 75.9 |
| Res-Tuning (Jiang et al., 2023) | 0.79 | 90.1 | 0.64 | 82.3 | 85.4 | 61.2 | 74.1 |
| Bi-LoRA (Jie et al., 2023) | 0.24 | 89.3 | 0.28 | 81.1 | 84.4 | 60.5 | 75.4 |
| SynQT (Zhang et al., 2024a) | 0.30 | 84.7 | 0.26 | 78.0 | 84.4 | 56.2 | 72.9 |
| PYRA (Xiong et al., 2024) | 0.34 | 86.2 | 0.30 | 79.1 | 84.4 | 60.6 | 74.7 |
| GPS (Zhang et al., 2024b) | 0.77 | 90.0 | 0.50 | **83.7** | 80.2 | **61.9** | 75.2 |
| Adapter+(r=1) (Steitz & Roth, 2024) | 0.22 | 90.1 | 0.23 | 83.2 | 85.5 | 60.1 | 76.3 |
| DyT (r=0.5) (Zhao et al., 2024) | 0.23 | 90.0 | 0.23 | 80.8 | 85.6 | 60.7 | 75.7 |
| *Our Methods (Dynamic)* | | | | | | | |
| Static DAF (Ours) | **0.21** | 89.5 | 0.22 | 81.5 | 85.2 | 60.8 | 75.8 |
| **DAF (Ours)** | **0.21** | **90.2** | 0.22 | 82.0 | **85.9** | 61.4 | **76.4** |

This is precisely the scenario where DAF's dynamic reconfiguration paradigm is designed to excel. By periodically re-evaluating and re-allocating trainable parameters, DAF can fluidly adapt to a model's evolving optimization priorities. The results in Table 2 provide strong evidence for this approach. DAF consistently outperforms the static baselines on both self-supervised backbones, suggesting its dynamic mechanism is uniquely suited to unlocking their full potential by effectively navigating their complex optimization landscapes.

Table 2: Performance comparison on the VTAB-1k benchmark across different ViT-B/16 self-supervised backbones.

| Backbone | Method | Natural | Specialized | Structured | Mean Acc. (%) |
|---|---|---|---|---|---|
| MAE | Full Fine-tuning | 59.3 | 79.7 | 53.8 | 64.3 |
| | LoRA-16 (Hu et al., 2022) | 57.3 | 77.1 | 59.9 | 64.8 |
| | Adapter-32 (Houlsby et al., 2019) | 55.3 | 78.8 | 53.3 | 62.5 |
| | SPT-Adapter (He et al., 2023) | 64.8 | 82.4 | 60.4 | 69.2 |
| | SPT-LoRA (He et al., 2023) | 63.8 | 81.6 | 60.0 | 68.5 |
| | BIAS (Zaken et al., 2022a) | 54.6 | 75.7 | 47.7 | 59.3 |
| | VPT-Deep (Jia et al., 2022) | 50.8 | 76.4 | 37.3 | 54.8 |
| | VQT (Tu et al., 2023) | 56.6 | 78.6 | 43.4 | 59.5 |
| | SynQT (Zhang et al., 2024a) | **66.0** | 82.6 | 58.2 | 68.9 |
| | **DAF (Ours)** | 65.7 | **82.7** | **61.4** | **69.9** |
| MoCo v3 | Full Fine-tuning | 72.0 | 84.7 | 42.0 | 69.6 |
| | LoRA-16 (Hu et al., 2022) | 16.0 | 64.0 | 48.7 | 42.9 |
| | Adapter-32 (Houlsby et al., 2019) | 74.2 | 82.7 | 47.7 | 68.2 |
| | SPT-Adapter (He et al., 2023) | 76.1 | 84.9 | 60.1 | 73.7 |
| | SPT-LoRA (He et al., 2023) | 76.5 | 85.4 | 63.0 | 75.0 |
| | BIAS (Zaken et al., 2022a) | 72.9 | 81.1 | 53.4 | 69.2 |
| | **DAF (Ours)** | **76.7** | **85.9** | **63.7** | **75.4** |

## 4.4 PERFORMANCE ON COMPLEX VISUAL TASKS

**Object Detection on MS COCO.** We utilized the MS COCO 2017 dataset (Lin et al., 2014) and employed Mask R-CNN (He et al., 2017) as the object detection framework. Following standard practices for prediction tasks, we adopted a Swin-B backbone initialized from ImageNet-21k weights. As reported in Table 3 (Left), DAF achieves a remarkable 53.5 $AP^{box}$ and 46.1 $AP^{mask}$. Notably, DAF surpasses the static LoRA baseline by a substantial margin of 3.1 $AP^{box}$ and 2.2 $AP^{mask}$, while using only 2.1M trainable parameters. This shows that our dynamic budget allocation effectively captures the multi-scale object-centric features essential for precise localization.

**Semantic Segmentation on ADE20K.** We further extended our evaluation to the ADE20K dataset (Zhou et al., 2017) using the UperNet (Xiao et al., 2018) framework with a stronger Swin-L

backbone.As shown in Table 3 (Right), DAF attains a best mIoU of 52.0%. This performance significantly outperforms the LoRA baseline (50.3%) by 1.7% and surpasses the previous best method LoRand++ with fewer parameters.

Table 3: Results on complex visual tasks. Left: Object Detection and Instance Segmentation on COCO val2017. Right: Semantic Segmentation on ADE20K. We include recent SOTA methods on Swin backbones.

| Object Detection (COCO) | | | | | Semantic Segmentation (ADE20K) | | | |
|---|---|---|---|---|---|---|---|---|
| **Method** | **Backbone** | **Params** | $\textbf{AP}^{box}$ | $\textbf{AP}^{mask}$ | **Method** | **Backbone** | **Params** | **mIoU** |
| Full Fine-tuning | Swin-B | 89M | 52.4 | 45.1 | Full Fine-tuning | Swin-L | 198M | 51.1 |
| *Recent SOTA baselines* | | | | | | | | |
| BitFit (Zaken et al., 2022b) | Swin-B | 0.2M | 50.1 | 43.6 | BitFit | Swin-L | 0.3M | 48.3 |
| NormTuning (Giannou et al., 2023) | Swin-B | 0.1M | 50.1 | 43.5 | NormTuning | Swin-L | 0.1M | 47.9 |
| Partial-1 (Yosinski et al., 2014) | Swin-B | 13.0M | 50.6 | 43.7 | Partial-1 | Swin-L | 28.8M | 47.4 |
| Adapter (Houlsby et al., 2019) | Swin-B | 3.2M | 52.1 | 45.0 | Adapter | Swin-L | 4.6M | 50.8 |
| LoRA (Hu et al., 2022) | Swin-B | 3.1M | 50.4 | 43.9 | LoRA | Swin-L | 4.6M | 50.3 |
| AdaptFormer (Chen et al., 2022) | Swin-B | 1.6M | 51.7 | 44.6 | AdaptFormer | Swin-L | 2.3M | 50.8 |
| LoRand (Yin et al., 2023b) | Swin-B | 2.4M | 51.1 | 44.1 | LoRand | Swin-L | 3.6M | 50.7 |
| LoRand++ (Yin et al., 2023b) | Swin-B | 9.3M | 51.5 | 44.4 | LoRand++ | Swin-L | 14.2M | 51.9 |
| Mona (Yin et al., 2024) | Swin-B | 4.2M | 53.4 | 46.0 | Mona | Swin-L | 5.1M | 51.4 |
| **DAF (Ours)** | Swin-B | 2.1M | **53.5** | **46.1** | **DAF (Ours)** | Swin-L | 3.7M | **52.0** |

These results align with observations in Mona (Yin et al., 2024) and LoRand (Yin et al., 2023b) that standard static PEFT methods often struggle with the complex spatial dependencies required in these tasks. By introducing the dynamic reconfiguration paradigm, DAF effectively bridges this gap. Additionally, we provide a comprehensive analysis of DAF's generalization capabilities on diverse backbones (e.g., ConvNeXt) for classification tasks in Appendix A.10, enabling flexible adaptation to diverse visual structures while maintaining superior parameter efficiency compared to both full fine-tuning and existing PEFT counterparts.

### 4.5 ABLATION STUDIES

To deeply understand and validate the contribution of each design choice within the DAF framework, we conducted a series of detailed ablation studies on the VTAB-1k benchmark.

**Impact of Core Components.** We first analyze the effectiveness of the core dynamic paradigm and its essential components, with results summarized in Table 4. (1) The comparison between the DAF framework and its static counterpart, Static DAF, directly validates the core hypothesis of this paper. Static DAF performs the context-aware decoupled analysis and rebuild-and-refocus but only once before training. By moving from this strong static baseline to a periodic reconfiguration strategy, DAF achieves a significant performance gain (76.4% vs 75.8%). To rigorously rule out the influence of hyperparameters, we further verify DAF against a wide spectrum of Static DAF configurations in Appendix A.9. This demonstrates that adapting to the model's evolving optimization priorities throughout the training process is crucial for unlocking higher performance under a constrained parameter budget. Having established the value of the dynamic paradigm, we further investigate the necessity of DAF's two key design choices. (2) We designed a variant, DAF-Naive, where the sensitivity analysis is always performed on a pristine, unchanged copy of the original ViT backbone, isolated from the model's evolving state. Its inferior performance highlights the criticality of the context-aware decoupled analysis. Making decisions based on the true, evolving state of the model provides a more accurate navigation signal, which is essential for effective dynamic adaptation. (3) We designed another variant, DAF-Accumulate, which adopts a pure knowledge accumulation approach where all historically activated LoRA modules are retained and remain trainable. This contrasts with the rebuild-and-refocus strategy, which freezes outdated modules. The superior performance of our main method shows that continuously training too many outdated modules wastes training resources and constrains the model's flexibility. The rebuild-and-refocus approach grants the model maximum agility by decisively reallocating its training budget to the most critical, evolving bottlenecks. In summary, these ablations compellingly demonstrate that DAF's success stems from a synergistic combination: the dynamic reconfiguration provides the opportunity for continuous improvement, while context-aware decoupled sensitivity analysis and the rebuild-and-refocus strategy provide the precise guidance and efficient execution necessary to realize that opportunity.

Table 4: Ablation study on the core components of DAF on the VTAB-1k benchmark.

| Method | Dynamic Reconfiguration | Context-Aware Analysis | Rebuild-and-Refocus | Mean Acc. (%) |
|---|---|---|---|---|
| Static DAF | | ✓ | ✓ | 75.8 |
| DAF-Naive | ✓ | | ✓ | 74.6 |
| DAF-Accumulate | ✓ | ✓ | | 75.3 |
| **DAF (Ours)** | ✓ | ✓ | ✓ | **76.4** |

**Impact of Dynamic Update Frequency and Budget.** We study the effect of the reconfiguration interval $E_{\text{interval}}$ and the parameter budget ratio $\tau$. As shown in Table 5 (left), updates that are too frequent (e.g., every 5 epochs) can lead to training instability, while updates that are too sparse (e.g., every 50 epochs) fail to capture the evolving training dynamics effectively. An interval of 10 epochs appears to offer the best trade-off. For the parameter budget $\tau$, shown in Table 5 (right), we observe that performance peaks at $\tau = 0.2$ and then slightly degrades as the budget continues to increase. This phenomenon can be explained by the nature of the sensitivity analysis. A relatively small subset of backbone parameters exhibits high sensitivity (i.e., large gradients), while the vast majority have very low sensitivity scores. A budget larger than optimal (e.g., $\tau = 0.3$ or $0.4$) forces the elite selection mechanism to include parameters with minimal sensitivity. Allocating training resources to these non-critical parameters is counterproductive; it can introduce noise into the optimization process and diverts the training budget away from the components that are most crucial for adaptation, leading to a marginal decline in performance. Therefore, $\tau = 0.2$ strikes an excellent balance, capturing a sufficient set of critical parameters for effective adaptation without wasting resources on less relevant ones.

Table 5: Impact of dynamic update frequency $E_{\text{interval}}$ (left) and parameter budget $\tau$ (right) on VTAB-1k average accuracy.

(a) Update frequency $E_{\text{interval}}$

| $E_{\text{interval}}$ (epochs) | 5 | 10 | 25 | 50 |
|---|---|---|---|---|
| Mean Acc. (%) | 76.0 | **76.4** | 76.1 | 75.9 |

(b) Parameter budget $\tau$

| $\tau$ | 0.1 | 0.2 | 0.3 | 0.4 |
|---|---|---|---|---|
| Mean Acc. (%) | 75.7 | **76.4** | 76.2 | 76.1 |

**Robustness to Sensitivity Batch Num.** We further investigated DAF's Robustness to sampling noise by varying the Sensitivity Batch Num $M$ (the number of batches used to estimate $s_p^{(t)}$). We swept $M$ across $\{8, 12, 16, 20, 28, 32\}$, with $M = 16$ being the default. As detailed in Appendix A.8, our results demonstrate high robustness: the Mean Accuracy on VTAB-1k fluctuates negligibly (within $\approx 0.3\%$) across this wide range. Furthermore, loss curves on CIFAR-100 confirm that training convergence remains smooth and consistent, validating that our context-aware analysis captures stable macro-level trends even with varying sample sizes.

## 5 CONCLUSION

In this paper, we propose a novel dynamic reconfiguration paradigm for PEFT and design a general framework named DAF to address the limitations of the static allocation paradigm prevalent in existing PEFT methods. Through a periodic perceive-decide-execute loop, DAF can continuously and adaptively reshape its fine-tuning structure based on the model's own learning state. The core contributions are threefold. First, the proposed dynamic reconfiguration paradigm challenges the static nature of existing methods. Second, we design a complete DAF framework, centered around a sophisticated Rebuild-and-Refocus update strategy. This strategy uniquely preserves learned knowledge in outdated modules by freezing them, while decisively refocusing the limited parameter budget on newly identified bottlenecks, thus maximizing adaptivity without catastrophic forgetting. Lastly, to provide precise guidance for dynamic decision-making, we pioneer a context-aware decoupled sensitivity analysis method. By temporarily freezing existing fine-tuning modules on the full model, this method elegantly resolves the signal noise problem in dynamic analysis. Extensive experiments on several challenging vision benchmarks compellingly show that our method not only significantly outperforms mainstream static PEFT baselines but also achieves SOTA performance.

Despite the encouraging results achieved by DAF, future work will explore combining the DAF approach with other PEFT techniques and extending it to broader domains such as multimodal learning, which opens a new path toward more intelligent adaptation of large-scale pretrained models.

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

# A APPENDIX

## A.1 DATASET DETAILS

The experiments are conducted on two diverse and challenging benchmarks: the Fine-Grained Visual Classification (FGVC) suite and the Visual Task Adaptation Benchmark (VTAB-1k) (Zhai et al., 2019).

The VTAB-1k benchmark consists of 19 distinct tasks as shown in Table 6, which are grouped into three categories. The Natural category includes common image classification datasets such as CIFAR-100 (Krizhevsky & Hinton, 2009), Caltech101 (Fei-Fei et al., 2004), DTD (Cimpoi et al., 2014), Oxford-Flowers102 (Nilsback & Zisserman, 2008), Oxford-Pets (Parkhi et al., 2012), SVHN (Netzer et al., 2011), and Sun397 (Xiao et al., 2010). The Specialized category is composed of tasks from specific domains, including medical imaging (Patch Camelyon (Veeling et al., 2018), Retinopathy (Kaggle and EyePacs, 2015)) and satellite imagery (EuroSAT (Helber et al., 2019), Resisc45 (Cheng et al., 2017)). The Structured category evaluates the model's understanding of scenes and semantics, with tasks like Clevr (Johnson et al., 2017), DMLab (Beattie et al., 2016), KITTI-Dist (Geiger et al., 2013), dSprites (Higgins et al., 2017), and SmallNORB (LeCun et al., 2004). For all VTAB-1k tasks, we follow the standard setup of using 800 training and 200 validation samples.

The FGVC benchmark focuses on tasks that require distinguishing between subtle visual differences. We evaluate on five datasets from this benchmark: CUB-200-2011 (Wah et al., 2011), NABirds (Van Horn et al., 2015), Oxford-Flowers102 (Nilsback & Zisserman, 2008), Stanford Cars (Krause et al., 2013), and Stanford Dogs (Khosla et al., 2011). For these datasets, we use the official training, validation, and test splits provided by the dataset creators.

Table 6: Statistics of the datasets used in the experiments. For VTAB-1k, all tasks use 800 training and 200 validation samples. For FGVC, we list the official split sizes.

| Benchmark | Dataset | # Classes | Train | Val | Test |
|---|---|---|---|---|---|
| **VTAB-1k** | | | | | |
| *Natural* | CIFAR100 | 100 | 800 | 200 | 10,000 |
| | Caltech101 | 102 | 800 | 200 | 6,084 |
| | DTD | 47 | 800 | 200 | 1,880 |
| | Oxford-Flowers102 | 102 | 800 | 200 | 6,149 |
| | Oxford-Pets | 37 | 800 | 200 | 3,669 |
| | SVHN | 10 | 800 | 200 | 26,032 |
| | Sun397 | 397 | 800 | 200 | 21,750 |
| *Specialized* | Patch Camelyon | 2 | 800 | 200 | 32,768 |
| | EuroSAT | 10 | 800 | 200 | 5,400 |
| | Resisc45 | 45 | 800 | 200 | 6,300 |
| | Retinopathy | 5 | 800 | 200 | 42,670 |
| *Structured* | Clevr/count | 8 | 800 | 200 | 15,000 |
| | Clevr/distance | 6 | 800 | 200 | 15,000 |
| | DMLab | 6 | 800 | 200 | 22,735 |
| | KITTI-Dist | 4 | 800 | 200 | 711 |
| | dSprites/location | 16 | 800 | 200 | 73,728 |
| | dSprites/orientation | 16 | 800 | 200 | 73,728 |
| | SmallNORB/azimuth | 18 | 800 | 200 | 12,150 |
| | SmallNORB/elevation | 18 | 800 | 200 | 12,150 |
| **FGVC** | | | | | |
| *FGVC* | CUB-200-2011 | 200 | 5,994 | – | 5,794 |
| | NABirds | 555 | 23,929 | – | 24,633 |
| | Oxford-Flowers102 | 102 | 1,020 | 1,020 | 6,149 |
| | Stanford Cars | 196 | 8,144 | – | 8,041 |
| | Stanford Dogs | 120 | 12,000 | – | 8,580 |

## A.2 MORE IMPLEMENTATION DETAILS

We provide a comprehensive list of hyperparameters used for training the DAF framework and all baselines in Table 7. These settings are kept consistent across all datasets to ensure a fair comparison, unless otherwise specified in the original papers of the baseline methods. All experiments were conducted using the PyTorch framework.

Table 7: General hyperparameters used for all experiments.

| Hyperparameter | Value |
|---|---|
| *Optimizer* | |
| Optimizer | AdamW |
| Betas | (0.9, 0.999) |
| Epsilon | $1 \times 10^{-8}$ |
| *Training Schedule* | |
| Base Learning Rate | $3 \times 10^{-4}$ |
| Weight Decay | $1 \times 10^{-4}$ |
| Learning Rate Schedule | Cosine Decay |
| Warmup Epochs | 10 |
| Batch Size | 64 |
| Total Epochs | 200 |
| *Regularization* | |
| Label Smoothing | 0.1 |
| Drop Path Rate | 0.1 |
| *DAF Specific* | |
| LoRA Rank ($r$) | 8 |
| Dynamic Update Interval ($E_{\text{interval}}$) | 10 |
| Parameter Budget ($\tau$) | 0.2 |
| Sensitivity Analysis Batches | 8 |
| Senssitivity Batch Num | 16 |

## A.3 VISUAL ANALYSIS OF TRAINING STABILITY

To empirically address the concern regarding training stability due to optimizer state re-initialization, we visualized the training loss trajectories over 200 epochs on four representative datasets: CIFAR-100, Caltech-101, DTD, and Flower102.

As shown in Figure 4, we observe the following key behaviors:

**Fast Convergence.** Across all datasets, DAF (Red line) demonstrates the fastest convergence rate, typically stabilizing around epoch 100-125. This indicates that the dynamic allocation of resources allows the model to fit the data more efficiently than static baselines.

**Realistic Dynamics.** The loss curves exhibit natural fluctuations (jitter), particularly for DAF. This behavior reflects the periodic "Perceive-Decide-Execute" process. Rather than being a sign of instability, these minor fluctuations represent the model actively escaping local minima and exploring better optimization paths, a mechanism akin to simulated annealing.

**Superior Final Loss.** Crucially, after converging (post-150 epochs), DAF consistently settles at a lower loss level than both Static DAF and LoRA. For instance, on CIFAR-100, the gap between DAF and LoRA is significant, aligning with the large accuracy gap (74.1% vs 68.1%).

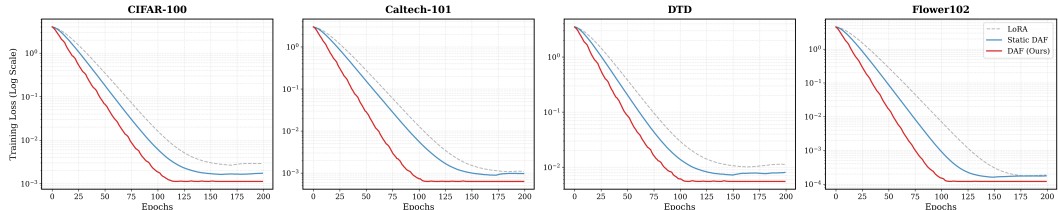

Figure 4: Training Loss. Comparison of DAF (Ours), Static DAF, and LoRA. DAF consistently achieves the fastest convergence and the lowest final loss. The slight jitter in the DAF curve reflects its dynamic nature, effectively preventing stagnation in sub-optimal local minima.

## A.4 DETAILED PER-TASK RESULTS

Here we provide a more granular analysis with detailed per-task results on both the FGVC and VTAB-1k benchmarks, presented in Table 8 and Table 9 respectively. In the fine-grained tasks of FGVC (Table 8), where identifying subtle visual cues is paramount, DAF's ability to shift its focus during training allows it to capture a wider range of discriminative features. On the diverse tasks of VTAB-1k (Table 9), DAF shows notable strength, particularly on the Structured category. These tasks (e.g., DMLab, KITTI-Dist) exhibit a large domain shift from the pre-training data, a scenario where static methods often falter. DAF's adaptability allows it to better navigate these challenging domain gaps, leading to superior robustness.

Table 8: Detailed per-task Top-1 accuracy (%) on the FGVC benchmark with ViT-B/16 (ImageNet-21k pre-trained).

| Method | CUB-200-2011 | NABirds | OxfordFlowers | StanfordDogs | StanfordCars | Mean Acc. (%) |
|---|---|---|---|---|---|---|
| Full Fine-tuning | 87.3 | 82.7 | 98.8 | 89.4 | 84.5 | 88.5 |
| *Static PEFT Baselines* | | | | | | |
| Adapter-8 | 87.3 | **84.3** | 98.4 | 88.8 | 68.4 | 85.5 |
| Adapter-32 | 87.2 | **84.3** | 98.5 | 89.6 | 68.4 | 85.6 |
| LoRA-8 | 84.9 | 79.0 | 98.1 | 88.1 | 79.8 | 86.0 |
| LoRA-16 | 85.6 | 79.8 | 98.9 | 87.6 | 72.0 | 84.8 |
| VPT-Deep | 84.5 | 78.5 | 98.5 | 86.5 | 71.0 | 83.8 |
| AdaptFormer | 87.0 | 83.2 | 98.8 | 89.0 | 72.5 | 86.1 |
| NOAH | 88.6 | 83.3 | 99.3 | 90.8 | 84.0 | 89.2 |
| *Recent Static SOTA Baselines* | | | | | | |
| VQT | 83.0 | 77.0 | 97.5 | 85.0 | 70.0 | 82.5 |
| SPT-LoRA | 88.6 | 82.8 | **99.4** | 91.4 | 84.5 | 89.3 |
| SPT-Adapter | **89.1** | 83.3 | 99.2 | 90.5 | 85.6 | 89.5 |
| Res-Tuning | 88.7 | 83.8 | 99.3 | 91.4 | 87.3 | 90.1 |
| Bi-LoRA | 87.8 | 83.5 | 99.0 | 90.2 | 86.0 | 89.3 |
| SynQT | 85.5 | 79.5 | 98.7 | 87.8 | 72.0 | 84.7 |
| PYRA | 86.8 | 82.5 | 98.8 | 89.1 | 73.8 | 86.2 |
| GPS | 88.6 | 83.6 | 99.3 | 91.3 | 87.2 | 90.0 |
| Adapter+(r=1) | 88.6 | 83.9 | 99.3 | 91.4 | 87.3 | 90.1 |
| DyT(r=0.5) | 88.5 | 83.8 | 99.3 | 91.3 | 87.1 | 90.0 |
| *Our Methods* | | | | | | |
| Static DAF (Ours) | 88.2 | 82.5 | 99.0 | 91.0 | 86.8 | 89.5 |
| **DAF (Ours)** | 88.7 | 83.9 | **99.4** | **91.5** | **87.4** | **90.2** |

Table 9: Detailed per-task Top-1 accuracy (%) on the VTAB-1k benchmark with ViT-B/16 (ImageNet-21k pre-trained). Note the performance across Natural, Specialized, and Structured task categories.

| Method | Natural | | | | | | | Specialized | | | | Structured | | | | | | | | Mean Acc. (%) |
|---|---|---|---|---|---|---|---|---|---|---|---|---|---|---|---|---|---|---|---|---|
| | CIFAR100 | Caltech101 | DTD | Flower102 | Pets | SVHN | Sun397 | EuroSAT | Resisc45 | Retinopathy | Camelyon | Clevr-Count | Clevr-Dist | DMLab | KITTI-Dist | dSpr-Loc | dSpr-Ori | sNORB-Azim | sNORB-Ele | |
| Full Fine-tuning | 68.9 | 87.7 | 64.3 | 97.2 | 86.9 | 87.4 | 38.8 | 95.7 | 84.2 | 73.9 | 79.7 | 56.3 | 58.6 | 41.7 | 65.5 | 57.5 | 46.7 | 25.7 | 29.1 | 69.0 |
| *Static PEFT Baselines* | | | | | | | | | | | | | | | | | | | | |
| Adapter-32 | 68.7 | 92.2 | 69.8 | 98.9 | 90.3 | 84.2 | 53.0 | 95.4 | 83.2 | 74.3 | 83.2 | 81.9 | 63.9 | 48.7 | 80.6 | 76.2 | 47.6 | 30.8 | 36.4 | 74.0 |
| LoRA-16 | 68.1 | 91.4 | 69.8 | 99.0 | 90.5 | 86.4 | 53.1 | 95.8 | 84.7 | 74.2 | 85.1 | 83.0 | 66.9 | 50.4 | 81.4 | 80.2 | 46.6 | 32.2 | 41.1 | 75.0 |
| VPT-Deep | 78.8 | 90.8 | 65.8 | 98.0 | 88.3 | 78.1 | 49.6 | 96.1 | 83.4 | 68.4 | 81.8 | 68.5 | 60.0 | 46.5 | 72.8 | 73.6 | 47.9 | 32.9 | 37.8 | 72.0 |
| AdaptFormer | 70.8 | 70.8 | 70.5 | 99.1 | 90.9 | 86.6 | 54.8 | 95.8 | 84.4 | 76.3 | 83.0 | 81.9 | 64.3 | 49.3 | 80.3 | 76.3 | 45.7 | 31.7 | 41.1 | 74.8 |
| NOAH | 69.6 | 92.7 | 70.2 | 99.1 | 90.4 | 86.1 | 53.7 | 95.4 | 83.9 | 75.8 | 84.4 | 82.8 | **68.9** | 49.9 | 81.7 | 81.8 | 48.3 | 32.8 | 44.2 | 75.5 |
| *Recent Static SOTA Baselines* | | | | | | | | | | | | | | | | | | | | |
| VQT | 66.3 | 89.9 | 67.8 | 97.9 | 84.7 | 79.9 | 45.5 | 95.2 | 80.9 | 74.7 | 79.0 | 46.7 | 61.6 | 45.1 | 63.6 | 62.9 | 32.1 | 30.0 | 28.8 | 68.3 |
| SPT-Adapter | 72.9 | 93.2 | 72.5 | 99.3 | 91.4 | 84.6 | 55.2 | 96.0 | 84.3 | 75.5 | 85.3 | 82.2 | 68.0 | 49.3 | 80.0 | 82.4 | 51.9 | 31.7 | 41.2 | 75.8 |
| SPT-LoRA | 72.3 | 93.0 | 72.5 | 99.3 | 91.5 | 86.2 | 55.5 | 96.2 | 85.1 | 75.9 | 85.0 | 83.7 | 66.4 | 52.5 | 80.2 | 80.1 | 51.1 | 30.1 | 41.3 | 75.9 |
| Res-Tuning | 75.2 | 92.7 | 71.9 | 99.3 | 91.9 | 86.7 | **58.5** | 95.6 | 85.0 | 74.6 | 86.7 | 80.2 | 63.6 | 50.6 | 80.2 | **85.4** | **55.7** | 31.9 | 42.0 | 74.10 |
| Bi-LoRA | 72.6 | 90.4 | 71.8 | 99.0 | 91.3 | 87.0 | 56.0 | 94.1 | 82.1 | 75.4 | 86.1 | 81.0 | 64.2 | 50.5 | 79.7 | 83.0 | 53.7 | 29.7 | 42.9 | 75.4 |
| SynQT | 70.9 | 89.7 | 68.8 | 98.5 | 89.6 | 77.8 | 50.6 | **96.7** | 83.5 | 75.2 | 82.3 | 71.8 | 62.7 | 48.5 | 75.4 | 74.1 | 49.0 | 31.7 | 36.1 | 72.9 |
| PYRA | 67.5 | 90.3 | 69.3 | 98.9 | 90.0 | 84.6 | 53.1 | 95.7 | 83.3 | 75.2 | 83.3 | 82.6 | **68.9** | 50.8 | 80.0 | 81.8 | 45.8 | 32.2 | 42.8 | 74.7 |
| GPS | 81.1 | **94.2** | 75.8 | **99.4** | 91.7 | 91.6 | 52.4 | 96.2 | **86.5** | **76.5** | **87.9** | 79.9 | 62.6 | **55.0** | **82.4** | 84.0 | 55.4 | 29.7 | **46.1** | 75.2 |
| Adapter(r=1) | **85.4** | 92.4 | 73.1 | 99.1 | 91.3 | 83.1 | 58.1 | 96.6 | 85.3 | 72.6 | 87.2 | 80.7 | 60.6 | 50.9 | 79.9 | 83.3 | 55.6 | 27.1 | 43.0 | 76.3 |
| DyT(r=0.5) | 70.4 | **94.2** | 71.1 | 99.1 | 91.7 | **88.0** | 51.5 | 95.3 | 84.2 | 75.8 | 87.1 | 79.2 | 61.8 | 51.0 | **82.4** | 79.7 | 52.3 | **35.3** | 44.5 | 75.7 |
| *Our Methods* | | | | | | | | | | | | | | | | | | | | |
| Static DAF (Ours) | 73.4 | 92.4 | 72.8 | 99.1 | 91.0 | 86.7 | 55.0 | 95.8 | 84.4 | 75.2 | 85.3 | 83.9 | 67.3 | 51.8 | 82.0 | 80.4 | 50.1 | 30.1 | 40.9 | 75.8 |
| **DAF (Ours)** | 74.1 | 92.9 | 73.0 | **99.4** | 91.7 | 87.5 | 55.5 | 96.1 | 85.6 | 75.8 | 86.0 | **84.5** | 67.8 | 52.5 | 82.1 | 81.2 | 50.9 | 30.1 | 41.4 | **76.4** |

## A.5 DYNAMIC BEHAVIOR VISUALIZATION ON VTAB-1K

To provide a comprehensive and intuitive understanding of DAF's working mechanism across diverse tasks, we visualize the evolution of activated LoRA modules during training for all 19 datasets in the VTAB-1k benchmark. As depicted in Figure 5, while the general pattern of shifting focus is consistent, the specific layers and the timing of these shifts vary significantly from task to task.

For instance, on semantically simple tasks like EuroSAT, the model quickly identifies and focuses on a stable set of features. In contrast, on complex, structured tasks like DMLab, the activated regions exhibit a much more volatile and continuous redistribution throughout the entire training process. This adaptive behavior, which no static PEFT method can replicate, visually demonstrates how DAF intelligently and uniquely allocates its limited parametric resources for each specific task, adapting to where they are most needed as the model's learning state evolves.

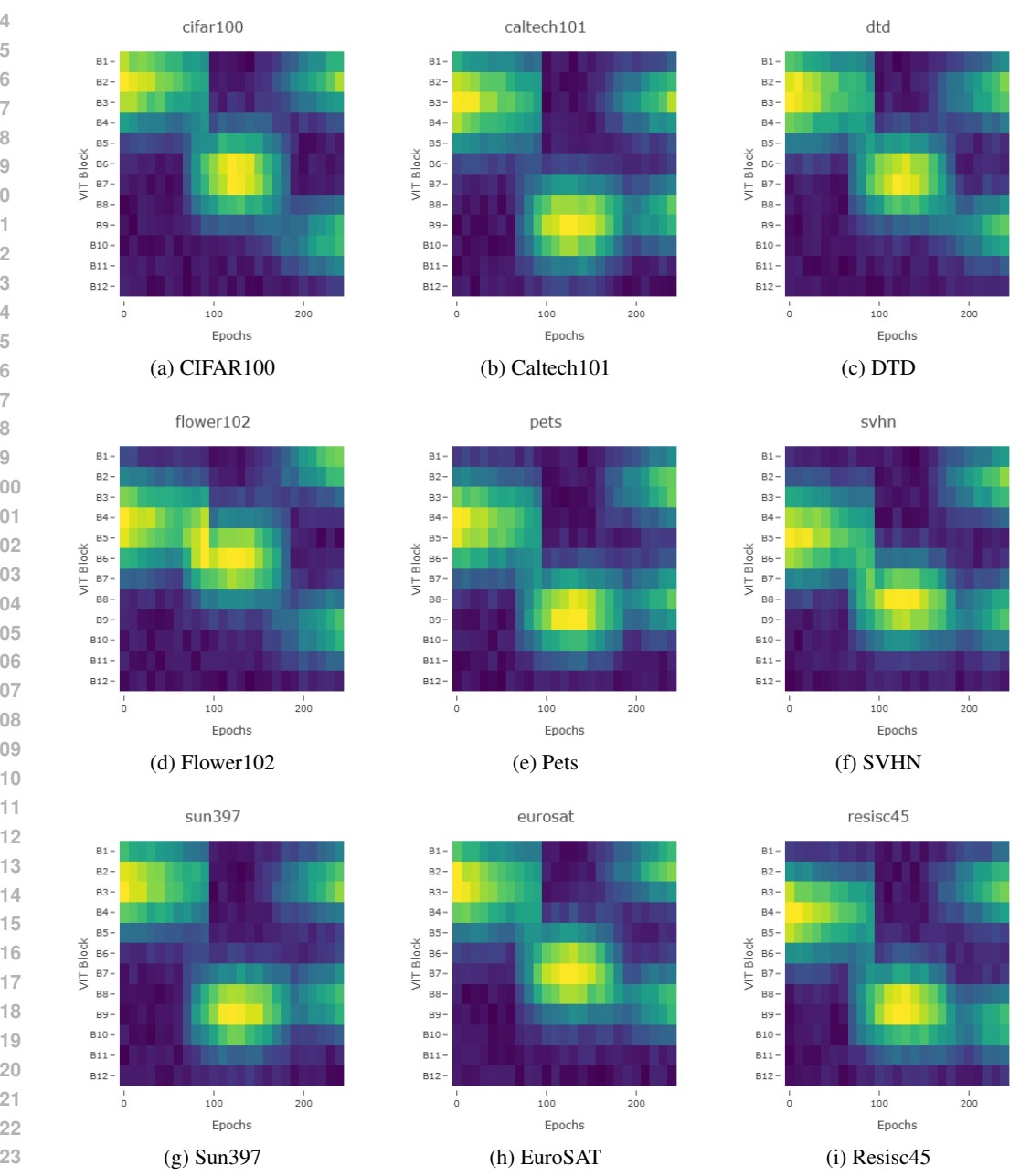

(a) CIFAR100         (b) Caltech101         (c) DTD

(d) Flower102         (e) Pets         (f) SVHN

(g) Sun397         (h) EuroSAT         (i) Resisc45

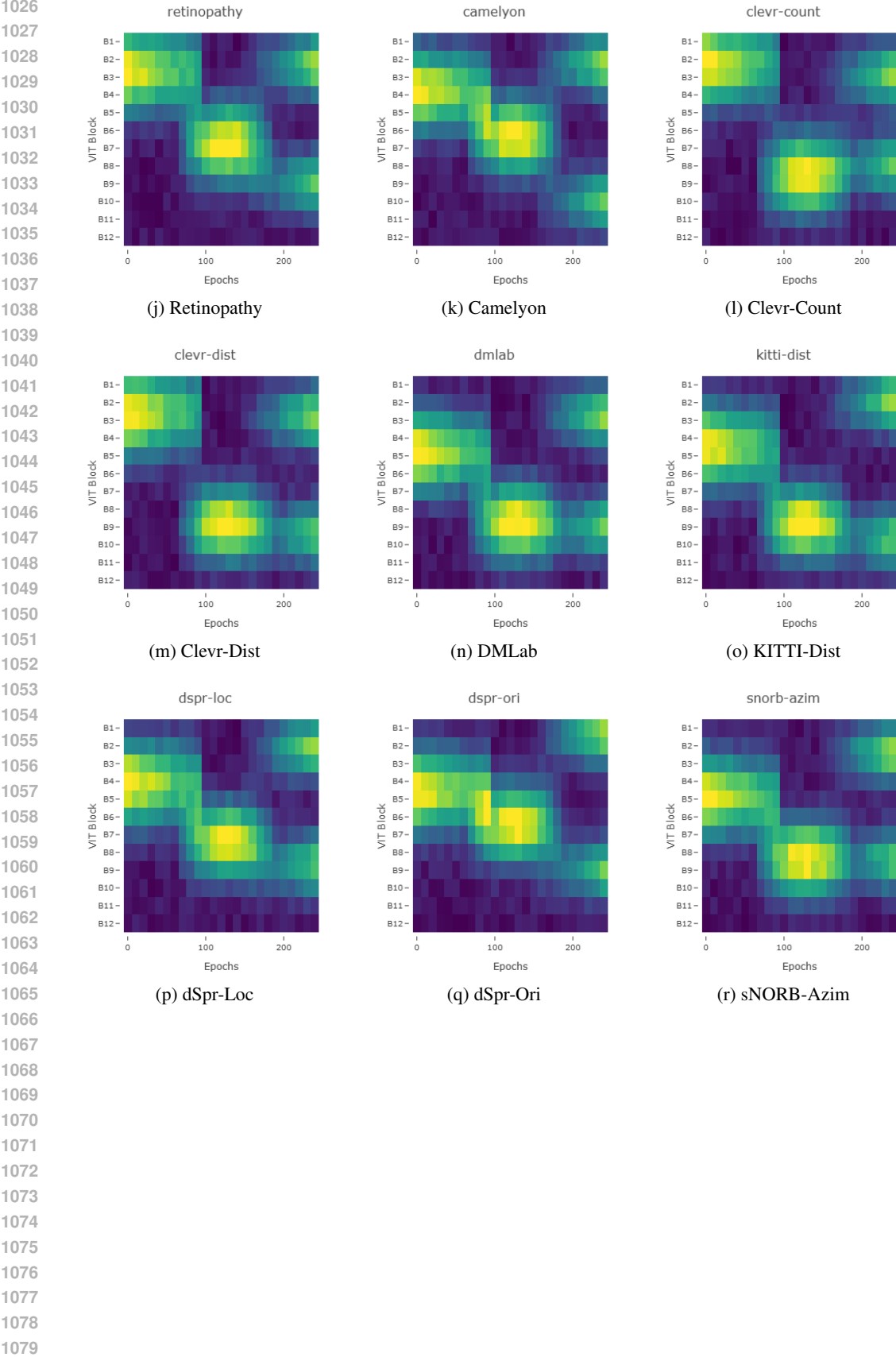

(j) Retinopathy

(k) Camelyon

(l) Clevr-Count

(m) Clevr-Dist

(n) DMLab

(o) KITTI-Dist

(p) dSpr-Loc

(q) dSpr-Ori

(r) sNORB-Azim

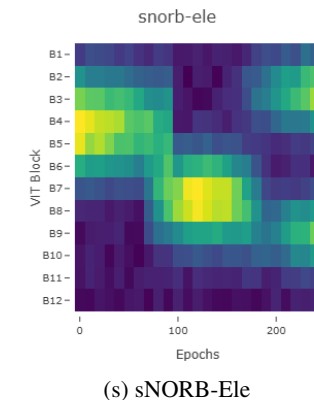

(s) sNORB-Ele

Figure 5: The dynamic evolution of activated modules by DAF across all 19 tasks in the VTAB-1k benchmark. The x-axis represents training epochs (up to 200), and the y-axis represents the 12 blocks of ViT. The color intensity indicates the activation level of LoRA modules. Note the diversity in activation patterns, demonstrating DAF's task-specific adaptability.

## A.6 COMPUTATIONAL OVERHEAD ANALYSIS

In this subsection, we provide a comparative analysis of the computational overhead of the DAF framework against other representative PEFT methods. We focus on three key efficiency metrics: the percentage of trainable parameters, the training memory footprint, and the inference computational cost (GFLOPs). A comprehensive comparison is presented in Table 10.

**Parameter and Memory Efficiency**. As shown in Table 10, DAF is highly parameter-efficient, requiring only 0.17M trainable parameters, which is the lowest among the compared methods. Furthermore, its training memory footprint of 8.64 GB is also highly efficient. While SynQT achieves the lowest memory usage due to its unique disentangled architecture that avoids back-propagation through the backbone, DAF's 8.64 GB represents a leading performance among methods that directly fine-tune the backbone weights, significantly outperforming standard LoRA-based approaches. This high efficiency is achieved because the periodic sensitivity analysis only momentarily increases memory usage, while the minimal set of active parameters keeps the optimizer state small throughout training. This contrasts with methods that require storing extensive intermediate features, which can lead to higher memory demands.

**Inference Efficiency**. A key advantage of DAF is its exceptional inference efficiency. Similar to LoRA and SPT-LoRA, the learned LoRA modules in DAF can be merged into the backbone weights before deployment. This re-parameterization means that DAF introduces zero additional inference latency or computational cost (GFLOPs) compared to the original, unmodified ViT backbone. This provides a distinct advantage over methods like VPT and SynQT. Although these methods are memory-efficient during training, they require separate modules during inference that cannot be merged. This results in increased computational cost for VPT (18.32 GFLOPs) and higher inference memory for SynQT (2.90 GB), making DAF a more streamlined solution for deployment.

In summary, DAF achieves a superior balance of parameter efficiency and deployment cost. The modest overhead introduced during the training phase by the dynamic reconfiguration mechanism is a strategic trade-off that unlocks significant performance gains (as shown in the main paper) without compromising the critical real-world deployment efficiency of the final model.

Table 10: Computational cost comparison on a ViT-B/16 backbone. Data for baseline methods are sourced from their respective publications on the VTAB-1k or similar benchmarks. DAF achieves a leading parameter efficiency while maintaining zero inference overhead.

| Method | Params (M) | Training Memory (GB) | Inference Memory (GB) | GFLOPs |
|---|---|---|---|---|
| Full Fine-tuning | 85.8 | 17.90 | 2.57 | 17.58 |
| Adapter | 1.19 | 13.74 | 2.61 | 17.81 |
| LoRA | 2.16 | 14.07 | 2.79 | 17.58 |
| AdaptFormer | 1.19 | 13.26 | 2.61 | 17.81 |
| VPT | 0.18 | 14.60 | 2.80 | 18.32 |
| SPT-Adapter | 0.35 | 13.50 | 2.61 | 17.81 |
| SPT-LoRA | 0.35 | 9.80 | **1.30** | 17.58 |
| PYRA | 0.29 | 13.30 | 2.61 | 17.81 |
| SynQT | 2.73 | **3.40** | 2.90 | **17.20** |
| NOAH | 0.45 | 13.50 | 2.61 | 17.81 |
| **DAF (Ours)** | **0.17** | 8.64 | 2.43 | 17.41 |

A.7    THE TRAINING AND INFERENCE TIMES FOR EACH MODULE OF DAF

In this section, we provide a quantitative breakdown of the computational time associated with the DAF framework, specifically focusing on the overhead introduced by the dynamic reconfiguration mechanism.

**Inference Efficiency: Zero Latency Increase.** A key design principle of DAF is to ensure efficient deployment. Although the model structure evolves during training, the final learned parameters (matrices $A$ and $B$) are mathematically merged into the backbone weights before inference:

$$W_{final} = W_0 + BA \tag{4}$$

This re-parameterization ensures that the deployed model is architecturally identical to the original ViT. Consequently, the inference time of DAF is exactly equal to that of the original pre-trained ViT model, introducing zero additional GFLOPs or latency.

**Training Efficiency: Stepwise Analysis.** To quantify the impact of the dynamic mechanism on training time, we profiled the DAF framework on the CIFAR-100 dataset using an NVIDIA RTX 4090 GPU. The training process alternates between a standard fine-tuning phase (lasting $E_{interval} = 10$ epochs) and a dynamic update phase. The time consumption for a single update cycle (10 epochs + 1 dynamic update) is detailed in Table 11.

Table 11: Time consumption breakdown for one DAF training cycle (10 epochs) on CIFAR-100.

| Phase | Component | Time Cost |
|---|---|---|
| **Standard Training** | Fine-tuning (10 epochs) | $8.12s \times 10$ $\approx$ **81.2s** |
| **Dynamic Update** | Context-Aware Analysis (Perceive) | $\approx 3.12s$ |
| | Elite Selection (Decide) | $\approx 0.21s$ |
| | Rebuild-and-Refocus (Execute) | $\approx 3.34s$ |
| | *Total Dynamic Overhead* | $\approx$ **6.67s** |

As illustrated in Table 11, the standard fine-tuning for 10 epochs consumes approximately 81.2 seconds. In contrast, the total overhead introduced by the dynamic operations sums to approximately 6.67 seconds. This represents a relative overhead of roughly 8.2% compared to the standard training time.

It is worth noting the efficiency difference between the dynamic stages. The Elite Selection (Decide) phase is extremely fast ($\approx 0.21s$) because it primarily involves sorting scalar sensitivity scores to identify the Top-$\tau$ parameters, which has very low computational complexity. Conversely, the Context-Aware Analysis (Perceive) takes longer ($\approx 3.12s$) as it necessitates a backward pass on the full model to compute accurate gradients. Overall, this minimal time cost validates that DAF achieves dynamic adaptability without imposing a heavy burden on training throughput. Furthermore, regarding inference, since the learned LoRA parameters are merged into the backbone weights via re-parameterization, DAF incurs zero additional computational costs during deployment, maintaining the exact same inference latency as the original ViT model

## A.8 ROBUSTNESS ANALYSIS OF SENSITIVITY ESTIMATION

To empirically validate the robustness of our sensitivity estimation against sampling noise, we conducted a comprehensive ablation study on the Sensitivity Batch Num $M$. We evaluated the DAF framework by varying $M$ from 8 to 32 on the VTAB-1k benchmark.

**Performance Stability.** As presented in Table 12, the final Mean Accuracy remains extremely stable across the entire range of $M \in \{8, 12, 16, 20, 28, 32\}$. The performance variance is minimal, indicating that the DAF framework is not sensitive to the exact number of batches used for sensitivity analysis, provided that a representative window is covered.

Table 12: Impact of Sensitivity Batch Num $M$ on VTAB-1k Mean Accuracy.

| Sensitivity Batch Num $M$ | 8 | 12 | 16 (Default) | 20 | 28 | 32 |
|---|---|---|---|---|---|---|
| Mean Acc. (%) | 76.3 | 76.4 | **76.4** | 76.5 | 76.4 | 76.2 |

**Training Convergence on CIFAR-100.** Furthermore, to visually assess the stability of the optimization process, we plotted the training loss curves for the CIFAR-100 task under different $M$ settings. As illustrated in Figure 6, the loss convergence remains smooth and consistent across all tested values ($M = 8$ to $M = 32$). We observed no signs of particular training instability or divergence, confirming that our multi-batch averaging strategy effectively mitigates the potential noise introduced by small-batch sampling.

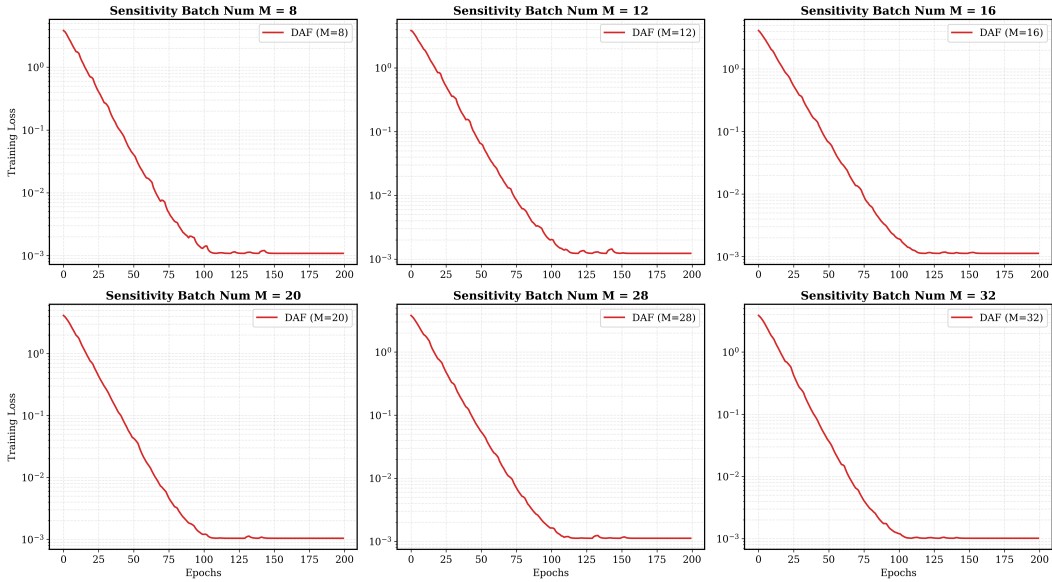

Figure 6: Training loss convergence curves on the CIFAR-100 dataset with varying Sensitivity Batch Num $M$ ($\{8, 12, 16, 20, 28, 32\}$). The overlapping curves exhibit consistent convergence behavior, demonstrating the stability of the dynamic reconfiguration process.

A.9 ROBUSTNESS ANALYSIS OF COMPARISON WITH STATIC DAF METHOD

To rigorously verify that the performance gain of DAF stems from its dynamic reconfiguration mechanism rather than suboptimal hyperparameter settings (e.g., learning rate or parameter budget) in the static baseline, we conducted a comprehensive spectrum analysis on the Static DAF.

**Experimental Setup.** We evaluated Static DAF across a broad grid of configurations, traversing different parameter budgets $\tau \in \{0.1, 0.2, 0.3\}$ and learning rates $lr \in \{1\text{e-}4, 3\text{e-}4, 5\text{e-}4\}$. This covers the potential optimal range for static fine-tuning.

**Results and Analysis.** The comparative results are visualized in Figure 7. We observe the following key findings:

- **Superiority of Dynamic Paradigm:** Regardless of the combination of learning rate and parameter budget, the performance ceiling of Static DAF consistently fails to surpass that of DAF. Even with the best-performing static configuration, a distinct gap remains compared to our dynamic method.

- **Optimization Interference in Static Allocation:** Simply increasing the parameter budget (e.g., from $\tau = 0.2$ to $0.3$) in the static setting does not yield continuous improvements and, in some cases, leads to performance degradation. This corroborates our hypothesis that activating non-critical parameters throughout the entire training process introduces optimization interference (gradient noise), which hinders the model's convergence.

These empirical results strongly demonstrate that the limitation of the baseline lies in its static assumption—the presumption that the elite parameters identified at initialization remain optimal throughout the training lifecycle. DAF breaks this assumption by dynamically shifting the budget to capture the temporal evolution of feature importance, thereby achieving superior adaptation efficiency that cannot be replicated by merely tuning static hyperparameters.

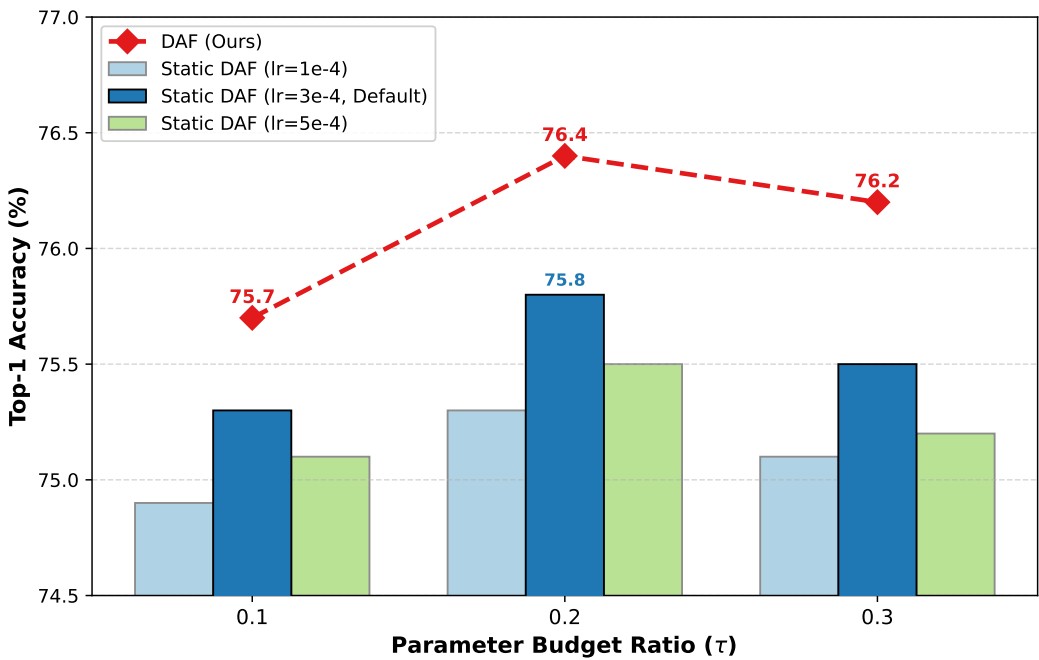

Figure 7: Performance comparison between DAF and Static DAF Spectrum on VTAB-1k. We compare DAF against Static DAF configured with various parameter budgets ($\tau$) and learning rates ($lr$). The red dashed line (or specific bar) represents DAF, which consistently outperforms all static configurations, validating the effectiveness of the dynamic reconfiguration paradigm.

## A.10 GENERALIZATION ON DIFFERENT ARCHITECTURES (SWIN & CONVNEXT)

To demonstrate the model-agnostic advantage of DAF, as recommended by the reviewers, we extended our evaluation to diverse architectures beyond standard ViTs. Specifically, we tested DAF on Swin-B (a hierarchical Transformer) and ConvNeXt-B (a modern CNN).

We conducted experiments on the FGVC benchmark (averaged accuracy over 5 fine-grained tasks: CUB-200, NABirds, Oxford Flowers, Stanford Dogs, and Stanford Cars). The pre-trained backbones were initialized from ImageNet-21k. For ConvNeXt, we adapted DAF by applying the dynamic selection mechanism to the pointwise convolutions ($1 \times 1$ convs, treating them as linear layers) and normalization layers.

The comparison with Full Fine-tuning, Linear Probing, SSF (Lian et al., 2022), and GPS (Zhang et al., 2024b) is presented in Table 13.

Table 13: Performance comparisons on the FGVC benchmark (Average accuracy over 5 tasks) with different model architectures. DAF achieves the best performance with the highest parameter efficiency on both Hierarchical Transformer and CNN backbones.

| Architecture | Swin-B | | ConvNeXt-B | |
|---|---|---|---|---|
| | Ave. Acc. | Params.(%) | Ave. Acc. | Params.(%) |
| Full | 92.42 | 100.00 | 93.04 | 100.00 |
| Linear | 87.90 | 0.28 | 88.00 | 0.28 |
| SSF | 91.54 | 0.56 | 92.48 | 0.56 |
| GPS | 92.56 | 0.95 | 93.32 | 0.90 |
| **DAF (Ours)** | **92.81** | **0.32** | **93.58** | **0.35** |

**Analysis of the Swin-B.** DAF outperforms GPS by +0.25% while using only 1/3 of the trainable parameters (0.32% vs. 0.95%). This confirms that DAF's dynamic reconfiguration is highly effective for hierarchical Transformer structures.

**Analysis of the ConvNeXt-B.** On the CNN-based architecture, DAF achieves a remarkable 93.58% average accuracy, surpassing both Full Fine-tuning and GPS. This indicates that our method successfully generalizes to CNNs, leveraging the dynamic sensitivity analysis to identify critical convolutional channels.

## A.11 THE USE OF LARGE LANGUAGE MODELS

During the preparation of this manuscript, a Large Language Model (LLM) is used as a writing assistant. Its role is strictly limited to improving grammar, phrasing, and overall readability. All scientific contributions, including the methodology and analysis of results, are solely performed by the authors.

