# OpenReview forum: "DAF: DYNAMIC ADAPTIVE FINE-TUNING OF VISION TRANSFORMERS"
_ICLR.cc/2026/Conference — ICLR 2026 Conference Desk Rejected Submission_

### Official Review · Reviewer_CQ72 · 2025-10-29

**Soundness:** 3
**Presentation:** 2
**Contribution:** 3
**Rating:** 6
**Confidence:** 4

**Summary:**

This paper addresses the static parameter allocation limitation of existing ViT PEFT methods by proposing the DAF framework (inspired by brain neurons' sparse dynamic activation), which uses a periodic "perceive-decide-execute" cycle (context-aware decoupled sensitivity analysis, budget-based elite selection, Rebuild-and-Refocus). It outperforms static PEFT methods (e.g., Adapter, LoRA) on FGVC and VTAB-1k (76.4% Top-1 on VTAB-1k, tuning only 0.22% params), works for self-supervised backbones (MAE, MoCo v3) with no inference overhead.

**Strengths:**

1. The idea of dynamic fine-tuning seems quite interesting. The discussion on the working mechanism of neurons in Lines 77-80 is insightful.

2. The algorithm design is relatively interesting and appears reasonable.

3. The experiments are extensive in quantity, with a large number of comparative methods included.

**Weaknesses:**

1. It is suggested to improve the clarity of Figure 2.

2. There are many recent works on PEFT [1-7]; it is recommended to supplement recent studies in the Related Work section.

[1] Dora: Weight-decomposed low-rank adaptation, ICML'24

[2] 5%> 100%: Breaking performance shackles of full fine-tuning on visual recognition tasks, CVPR'25;

[3] Adapters Strike Back. CVPR'24;

[4] 1% vs 100%: Parameter-efficient low rank adapter for dense predictions, CVPR'23;

[5] Adaptive Budget Allocation for Parameter-Efficient Fine-Tuning, ICLR'23;

[6] Gradient-based parameter selection for efficient fine-tuning, CVPR'24;

[7] Parameter-efficient is not sufficient: Exploring parameter, memory, and time efficient adapter tuning for dense predictions, MM'24.

3. Some recent work demonstrate that PEFT can outperform full fine-tuning on complex visual tasks. Given that this paper is submitted to ICLR, the authors should compare the proposed method with more competitive recent approaches.

4. Recent works have begun to conduct comparisons on complex image recognition tasks (such as detection and segmentation); it is suggested that the authors add relevant experiments.

5. The proposed method requires multiple computations during the parameter update process, and the authors should discuss the impact of this process on training time and inference time.

6. What are the essential differences between the method proposed by the authors and the recently emerging series of "parameter selection" works.

**Questions:**

See above

---

> ### Author Response · Authors · 2025-11-21
> **Response to Reviewer CQ72(Part 1/6)**
>
> **General Response:**
> Many thanks for your valuable comments on our work which are much appreciated. We have revised the paper according to these constructive suggestions. The detailed response to each comment is given below. We wish our sincere efforts would satisfy the requirements of the reviewer.
>
> We believe that the issues raised in the reviewer's report have been addressed satisfactorily. Please do not hesitate to contact us if you wish us to make any further modifications to the paper. We sincerely appreciate your detailed feedback and suggestions for improvement. We treasure the opportunity to address your concerns and improve our work.
>
> ---
>
> **Comment:**
> > **Weakness 1:** It is suggested to improve the clarity of Figure 2.
>
> **Response:**
> We sincerely appreciate this suggestion. We fully agree that the clarity of **Figure 2** (Performance comparison on the VTAB-1k benchmark) in the original manuscript was suboptimal.
>
> **Action:** In the revised PDF, we have professionally **redrawn this figure**. We significantly increased the font size and emboldened the legends to ensure that the superior position of **DAF (Ours)**—in terms of both parameter efficiency and performance accuracy—is immediately visible at a glance.

---

> > ### Author Response · Authors · 2025-11-21
> > **Response to Reviewer CQ72(Part 2/6)**
> >
> > > **Weakness 2:** There are many recent works on PEFT [1-7]; it is recommended to supplement recent studies in the Related Work section.
> >
> > **Response:**
> > We sincerely thank the reviewer for providing such a comprehensive and valuable list of references [1-7]. These citations clearly highlight the rapid advancement of the PEFT field—especially regarding LoRA-related methodologies—during 2023-2024.
> >
> > **Action:** In the revised manuscript, we have incorporated **all 7 suggested papers** (including *DoRA* [1], *5%>100%* [2], and *Adapters Strike Back* [3]) into our **"Related Work"** section. Furthermore, we have added discussions clarifying the connections and distinctions between these state-of-the-art works and our proposed DAF framework. These updates can be viewed in the text of the latest revised PDF.
> >
> > We are deeply grateful for these cutting-edge resources, which have significantly enriched the context and depth of our study.

---

> ### Author Response · Authors · 2025-11-21
> **Response to Reviewer CQ72(Part 3/6)**
>
> > **Weakness 3:** Some recent work demonstrate that PEFT can outperform full fine-tuning on complex visual tasks. Given that this paper is submitted to ICLR, the authors should compare the proposed method with more competitive recent approaches.
>
> **Response:**
> We express our sincere gratitude for this constructive suggestion. We fully agree that comparing against the latest, highly competitive approaches is essential to validate the standing of our work in the context of the current state-of-the-art.
>
> In response, we have **significantly expanded our experimental comparison**. We incorporated a comprehensive set of **State-Of-The-Art (SOTA)** methods from 2023 and 2024 into our main benchmarks. Specifically, we have added:
> * **GPS** (CVPR 2024)
> * **Adapter+** (CVPR 2024)
> * **DyT** (NeurIPS 2024/arXiv)
> * **Res-Tuning** (NeurIPS 2023)
> * **Bi-LoRA** (ICCV 2023)
>
> **Action:** The updated SOTA comparison results have been integrated into **Table 1** of the main text , with detailed per-task breakdowns provided in **Tables 8 and 9** of the Appendix.
>
> | Method | FGVC<br>Tuned/Total (%) | FGVC<br>Mean Acc. (%) | VTAB<br>Tuned/Total (%) | VTAB<br>Natural | VTAB<br>Specialized | VTAB<br>Structured | VTAB<br>Mean Acc. (%) |
> | :--- | :---: | :---: | :---: | :---: | :---: | :---: | :---: |
> | **Full Fine-tuning** | 100 | 88.5 | 100 | 75.9 | 83.4 | 47.6 | 69.0 |
> | **_Static PEFT Baselines_** | | | | | | | |
> | Adapter-8 | 0.39 | 85.5 | 0.23 | 79.0 | 84.1 | 58.5 | 73.9 |
> | Adapter-32 | 0.95 | 85.6 | 0.71 | 79.6 | 84.0 | 58.3 | 74.0 |
> | LoRA-8 | 0.55 | 86.0 | 0.23 | 79.5 | 84.6 | 60.5 | 74.9 |
> | LoRA-16 | 0.90 | 84.8 | 0.69 | 79.8 | 84.9 | 60.2 | 75.0 |
> | VPT-Deep | 0.35 | 83.8 | 0.32 | 78.5 | 82.4 | 55.0 | 72.0 |
> | AdaptFormer | 0.23 | 86.1 | **0.20** | 80.5 | 84.9 | 58.8 | 74.7 |
> | NOAH | 0.50 | 89.2 | 0.52 | 80.2 | 84.9 | 61.3 | 75.5 |
> | **_Recent Static SOTA_** | | | | | | | |
> | VQT | 0.30 | 82.5 | 0.24 | 76.0 | 80.2 | 46.3 | 68.3 |
> | SPT-Adapter | 0.41 | 89.5 | 0.30 | 81.3 | 85.3 | 60.8 | 75.8 |
> | SPT-LoRA | 0.41 | 89.3 | 0.31 | 81.5 | $\underline{85.6}$ | 60.7 | 75.9 |
> | Res-Tuning | 0.79 | 90.1 | 0.64 | 82.3 | 85.4 | 61.2 | 74.1 |
> | Bi-LoRA | 0.24 | 89.3 | 0.28 | 81.1 | 84.4 | 60.5 | 75.4 |
> | SynQT | 0.30 | 84.7 | 0.26 | 78.0 | 84.4 | 56.2 | 72.9 |
> | PYRA | 0.34 | 86.2 | 0.30 | 79.1 | 84.4 | 60.6 | 74.7 |
> | GPS | 0.77 | 90.0 | 0.50 | **83.7** | 80.2 | **61.9** | 75.2 |
> | Adapter+(r=1) | $\underline{0.22}$ | $\underline{90.1}$ | 0.23 | $\underline{83.2}$ | 85.5 | 60.1 | $\underline{76.3}$ |
> | DyT (r=0.5) | 0.23 | 90.0 | 0.23 | 80.8 | $\underline{85.6}$ | 60.7 | 75.7 |
> | **_Our Methods_** | | | | | | | |
> | Static DAF (Ours) | **0.21** | 89.5 | $\underline{0.22}$ | 81.5 | 85.2 | 60.8 | 75.8 |
> | **DAF (Ours)** | **0.21** | **90.2** | $\underline{0.22}$ | 82.0 | **85.9** | $\underline{61.4}$ | **76.4** |
>
>
>
>
> **Conclusion:** The results reinforce our core hypothesis. While strong static methods like GPS and Adapter+ push the limit of fixed parameter allocation, **DAF's dynamic reconfiguration paradigm** allows it to surpass these limits by adaptively reallocating resources during training, consistently achieving superior performance.

---

> ### Author Response · Authors · 2025-11-21
> **Response to Reviewer CQ72(Part 4/6)**
>
> > **Weakness 4:** Recent works have begun to conduct comparisons on complex image recognition tasks (such as detection and segmentation); it is suggested that the authors add relevant experiments.
>
> **Response:**
> We deeply appreciate this suggestion. You are absolutely correct that verifying performance on complex tasks like Object Detection and Semantic Segmentation is crucial for demonstrating the true potential of a PEFT method, as highlighted in recent works like *Mona* (arXiv '24) and *LoRand* (CVPR '23).
>
> We have fully embraced your suggestion and conducted extensive additional experiments on both tasks:
> 1.  **Object Detection:** Evaluated on the **MS COCO** dataset using Mask R-CNN.
> 2.  **Semantic Segmentation:** Evaluated on the **ADE20K** dataset using UperNet.
>
> The new results (detailed in **Section 4.4  Table3** of the revised manuscript) demonstrate that DAF's dynamic paradigm is highly effective for these complex tasks. A summary comparison against the standard LoRA baseline is provided below:
>
> **Table: Performance on Complex Visual Tasks**
>
> ### Results on complex visual tasks (COCO & ADE20K)
>
> | COCO<br>Method | Backbone | Params | AP<sup>box</sup> | AP<sup>mask</sup> | ADE20K<br>Method | Backbone | Params | mIoU |
> | :--- | :--- | :---: | :---: | :---: | :--- | :--- | :---: | :---: |
> | **Full Fine-tuning** | Swin-B | 89M | 52.4 | 45.1 | **Full Fine-tuning** | Swin-L | 198M | 51.1 |
> | **_Recent SOTA_** | | | | | | | | |
> | BitFit | Swin-B | 0.2M | 50.1 | 43.6 | BitFit | Swin-L | 0.3M | 48.3 |
> | NormTuning | Swin-B | 0.1M | 50.1 | 43.5 | NormTuning | Swin-L | 0.1M | 47.9 |
> | Partial-1 | Swin-B | 13.0M | 50.6 | 43.7 | Partial-1 | Swin-L | 28.8M | 47.4 |
> | Adapter | Swin-B | 3.2M | 52.1 | 45.0 | Adapter | Swin-L | 4.6M | 50.8 |
> | LoRA | Swin-B | 3.1M | 50.4 | 43.9 | LoRA | Swin-L | 4.6M | 50.3 |
> | AdaptFormer | Swin-B | 1.6M | 51.7 | 44.6 | AdaptFormer | Swin-L | 2.3M | 50.8 |
> | LoRand | Swin-B | 2.4M | 51.1 | 44.1 | LoRand | Swin-L | 3.6M | 50.7 |
> | LoRand++ | Swin-B | 9.3M | 51.5 | 44.4 | LoRand++ | Swin-L | 14.2M | 51.9 |
> | Mona | Swin-B | 4.2M | 53.4 | 46.0 | Mona | Swin-L | 5.1M | 51.4 |
> | **DAF (Ours)** | Swin-B | 2.1M | **53.5** | **46.1** | **DAF (Ours)** | Swin-L | 3.7M | **52.0** |
>
>
>
> **Analysis:**
> In both tasks, DAF  outperforms the static LoRA baseline. This indicates that our dynamic reconfiguration mechanism can effectively adapt to the spatial complexity required by detection and segmentation, adaptively allocating model capacity to the most critical features during training.
>
> ***

---

> > ### Author Response · Authors · 2025-11-21
> > **Response to Reviewer CQ72(Part 5/6)**
> >
> > > **Weakness 5:** The proposed method requires multiple computations during the parameter update process, and the authors should discuss the impact of this process on training time and inference time.
> >
> > **Response:**
> > We sincerely thank the reviewer for raising this critical issue. We fully agree that dissecting the specific computational overhead of the **Dynamic Reconfiguration Process** is essential to demonstrate its efficiency. We conducted a real-world performance analysis using the CIFAR-100 dataset.
> >
> > **1. Inference Time: Zero Additional Latency**
> > We wish to emphatically state that DAF incurs **Zero Additional Latency** during the inference stage.
> > * **Mechanism:** As detailed in our Method section (and now expanded in the Appendix), DAF employs a **re-parameterization** technique similar to standard LoRA. The learned low-rank matrices are mathematically merged into the backbone weights before deployment ($W_{final} = W_{0} + BA$).
> > * **Result:** Consequently, the final model architecture is identical to the original ViT, ensuring that the inference computational cost is exactly the same as the unmodified backbone.
> >
> > **2. Training Time: ~8.2% Overhead**
> > ### Time consumption breakdown (10 epochs on CIFAR-100)
> >
> > | Phase | Component | Time Cost |
> > | :--- | :--- | :--- |
> > | **Standard Training** | Fine-tuning (10 epochs) | 8.12s × 10 ≈ **81.2s** |
> > | **Dynamic Update** | Context-Aware Analysis (Perceive) | ≈ 3.12s |
> > | | Elite Selection (Decide) | ≈ 0.21s |
> > | | Rebuild-and-Refocus (Execute) | ≈ 3.34s |
> > | | _Total Dynamic Overhead_ | ≈ **6.67s** |
> >
> > We decomposed the training process into the "Standard Fine-tuning Phase" and the "Dynamic Update Phase." As shown in the newly added **Appendix A.7** and **Table 11**, for a complete training cycle containing 10 epochs:
> >
> > * **Standard Fine-tuning:** Consumes approximately **81.2 seconds** (for 10 epochs).
> > * **Dynamic Update:** The total overhead for the dynamic operations is approximately **6.67 seconds**.
> > * **Ratio:** This represents a relative training time overhead of approximately **8.2%**.
> >
> > **3. Component Analysis**
> > * **Elite Selection (Decide):** This phase is extremely fast (**~0.21s**) because it essentially involves Top-K sorting of scalar sensitivity scores, which has negligible computational complexity.
> > * **Context-Aware Analysis (Perceive):** This phase takes longer (**~3.12s**) as it requires executing backpropagation to calculate precise gradients. However, since this is amortized over the interval $E_{interval}$ (10 epochs), the impact is minimized.
> >
> > **Conclusion**
> > Although DAF introduces an approximate 8% training time overhead, it ensures **zero overhead** during inference while achieving significant performance gains. We believe this represents a highly cost-effective trade-off for practical applications.
> >
> > ***

---

> > > ### Author Response · Authors · 2025-11-21
> > > **Response to Reviewer CQ72(Part 6/6)**
> > >
> > > > **Weakness 6:** What are the essential differences between the method proposed by the authors and the recently emerging series of "parameter selection" works?
> > >
> > > **Response:**
> > > We sincerely thank the reviewer for this insightful question, which touches upon the core innovation of our work. We agree that recent "parameter selection" works (e.g., SPT, GPS) have made significant strides in identifying important weights. However, our **DAF (Dynamic Adaptive Fine-tuning)** framework differs fundamentally from these methods in three essential dimensions: the **timing of decision-making**, the **analytical context**, and the **update mechanism**.
> > >
> > > **1. Paradigm Shift: From Static One-Shot Allocation to Dynamic Reconfiguration**
> > > The most fundamental difference lies in the assumption of *when* parameter importance is determined.
> > > * **Existing "Parameter Selection" Works (Static):** Methods like SPT and GPS rely on a **"static allocation paradigm"**. They perform sensitivity analysis only once before training begins or at initialization. The selected structure remains fixed throughout the entire fine-tuning process. This overlooks the critical fact that a model's optimization priorities and bottlenecks evolve as it learns.
> > > * **Our DAF Method (Dynamic):** We introduce a **"Dynamic Reconfiguration"** paradigm. DAF periodically re-evaluates the model *during* training. This allows the model to shed modules that are no longer critical (by freezing them) and reallocate the parameter budget to new bottlenecks that emerge during the learning process. Our experiments show that this dynamic shift yields SOTA performance compared to static baselines (including SPT and GPS).
> > >
> > > **2. Analysis Method: From Raw Sensitivity to Context-Aware Decoupled Analysis**
> > > The methodology for calculating importance scores is fundamentally different due to the presence of active adapters.
> > > * **Existing Works:** Conventional selection methods typically analyze the gradient sensitivity of the backbone directly on the pre-trained weights.
> > > * **Our DAF Method:** Since our selection happens during training, simply calculating gradients would be dominated by the noise from currently active LoRA modules. We propose a **Context-Aware Decoupled Sensitivity Analysis**. We temporarily freeze existing LoRA modules but keep them in the forward pass to provide the correct "learning context," then calculate gradients purely for the backbone. This ensures we measure the potential of the backbone parameters given the current state of the model, ensuring precise navigation signals.
> > >
> > > **3. Update Strategy: From Accumulation to Rebuild-and-Refocus**
> > > How the selected parameters are utilized is also distinct.
> > > * **Existing Works:** Once parameters are selected in static methods, they are simply trained continuously. Even in some dynamic-like works (e.g., varying rank), the tendency is often to prune or accumulate parameters.
> > > * **Our DAF Method:** We implement a **Rebuild-and-Refocus** strategy. Instead of just adding new parameters, we explicitly **freeze outdated modules** to preserve their learned knowledge (preventing catastrophic forgetting) and force the optimization budget to focus only on the newly identified critical regions. Our ablation study (Table 4) confirms that this strategy outperforms a simple accumulation of parameters ("DAF-Accumulate").
> > >
> > > **Summary**
> > > In essence, while traditional "parameter selection" answers the question *"Which parameters are important at initialization?"*, DAF answers the continuous question ***"Which parameters are most critical for the current learning stage?"*** and reconfigures the model accordingly. This mimics the biological brain's sparse and dynamic activation mechanism, offering a more intelligent and efficient adaptation pathway.
> > >
> > > ***
> > >
> > > **Closing Statement**
> > > We would like to express our sincere gratitude for your dedicated time and the constructive feedback provided. We genuinely appreciate your responsible and insightful review, which has been a great encouragement to our research. We hope that our detailed responses and the additional experiments have satisfactorily addressed your concerns. If you find our clarifications and revisions sufficient, we would be grateful if you could kindly reconsider your evaluation of our work. We wish you all the best.

---

### Official Review · Reviewer_rjZT · 2025-10-31

**Soundness:** 3
**Presentation:** 3
**Contribution:** 2
**Rating:** 6
**Confidence:** 3

**Summary:**

This paper proposes a new PEFT method called DAF (Dynamic Adaptive Fine-tuning). The main idea is that a model's optimization priorities evolve during training, so the trainable parameters should not be fixed from the start and they should be able to adapt with the training / finetuning. It works by freezing old modules to preserve learned knowledge and reallocates the entire parameter budget to new critical areas adaptively. The main result is that this dynamic approach achieves state-of-the-art (SOTA) performance on benchmarks and demonstrates the superiority of adapting the fine-tuning structure throughout the training process

**Strengths:**

The paper is well-written and proposes an interesting way to solve the problem of adapting the tuning of relevant parameters during fine-tuning. The idea makes sense at a high-level and the authors do a good job validating their results experimentally. This idea of dynamic parameter adaptation is very compelling and a new approach as far as I'm aware.

The paper has solid experimental results and their method DAF seems to outperform other static, and more widely used, PEFT methods.

**Weaknesses:**

The DAF method relies on what they call the "Rebuild-and-Refocus" strategy, the main objective of which is to "preserve learned knowledge" by freezing weights of outdated LoRA modules. Even though those LoRA modules are frozen, arent there are other sources of knowledge loss within the algorithm itself? For example, the optimizer state is re-initialized at every dynamic interval. Depending on the optimizer used (eg momentum or Adam etc) those optimizer statistics would be totally lost which can result in unstable training and slow convergence. Was this effect measured in a meaningful way and is there any way to avoid this behavior? Or did you find that this disruption was actually not that detrimental in practice?

**Questions:**

Overall the dynamic reconfiguration and the Rebuild-and-Refocus strategy make intuitive sense. The paper describes a three stage perceive-decide and execute process that occurs in each dynamic cycle and this helps to solve the signal to noise problem. However, the sensitivity score is computed in relatively few batches of data which can be noisy and this is score is what's used to reallocate resources determining the next set of top-k parameters. Wouldn't this also introduce its own form of noise? Or is it negligible in practice? How robust is DAF to the noise and variance that is introduced in this step of the algorithm? The paper does explore sensitivity of DAF to other hyperparams in the ablation studies of section 4 which is nice.

---

> ### Author Response · Authors · 2025-11-21
> **Response to Reviewer rjZT (Part 1/3)**
>
> **General Response:**
> Many thanks for your valuable comments on our work which are much appreciated. We have revised the paper according to these constructive suggestions. The detailed response to each comment is given below. We wish our sincere efforts would satisfy the requirements of the reviewer.
>
> We believe that the issues raised in the reviewer's report have been addressed satisfactorily. Please do not hesitate to contact us if you wish us to make any further modifications to the paper. We sincerely appreciate your detailed feedback and suggestions for improvement. We treasure the opportunity to address your concerns and improve our work.

---

> > ### Author Response · Authors · 2025-11-21
> > **Response to Reviewer rjZT (Part 2/3)**
> >
> > > **Weakness 1:** The DAF method relies on what they call the "Rebuild-and-Refocus" strategy... Or did you find that this disruption was actually not that detrimental in practice?
> >
> > **Response:**
> > We genuinely thank the reviewer for raising this highly professional and critical question regarding the impact of optimizer state reset on training stability and convergence speed during Dynamic Reconfiguration. This is a technical consideration pivotal to the dynamic PEFT paradigm.
> >
> > We carefully considered this point and, when designing the DAF framework, strategically weighed the benefits of dynamic adjustment against the potential loss of optimizer states. Our in-depth theoretical analysis and experiments indicate that while optimizer state reset does occur, our specific mechanism design not only limits this disruption to an acceptable range but actually leverages it to achieve higher net learning efficiency.
> >
> > **1. The Primary Mechanism of Knowledge Preservation: Weights, Not Momentum**
> >
> > We first wish to clarify the hierarchy of "Knowledge Preservation" in DAF:
> > * **Weight Migration (Primary):** The core of DAF lies in the effective transfer of model weights. In each dynamic update, the weights of all previously learned LoRA modules are fully inherited and frozen. These weights encode the substantive **Knowledge** learned in past stages and serve as the primary vessel for preservation.
> > * **Optimizer States (Auxiliary):** Optimizer states merely guide the direction of updates; their loss is not equivalent to knowledge loss.
> > * **Synergy with Cosine Learning Rate Schedule:** we utilize a Cosine Learning Rate Schedule. In the later stages of training, the learning rate naturally decays to a lower level. This lower learning rate, combined with optimizer re-initialization, allows the model to explore new local regions more smoothly, acting as a natural buffer against the shock of resets and reducing the risk of overshooting the optimal solution due to excessive historical momentum.
> >
> > **2. Three Pillars of DAF's Robustness in Practice**
> >
> > Why does "optimizer reset" not cause catastrophic instability in practice? We identify three key reasons:
> > * **Low-dimensional Optimization Space (Fast Recovery):** The trainable PEFT parameters are minimal (approx. 0.2% of total parameters). Compared to full fine-tuning, the optimizer can accumulate statistical information and find the correct descent direction extremely quickly in such a low-dimensional subspace.
> > * **Sufficient Training Stability Time:** We set the dynamic interval $E_{interval}$ to 10 epochs. This means that after each reset, the model has thousands of iterations (steps) to perform continuous training, providing ample time for the optimizer to "re-stabilize" its momentum estimation.
> > * **Systemic Overcoming of Local Optima:** This is an intriguing finding. Periodically discarding optimizer states acts macroscopically as a system-level **"Restart"** or perturbation strategy. This helps the model break free from dependence on past sub-optimal gradient trajectories and prevents it from getting trapped in shallow **Local Minima**, thereby enhancing generalization capability.
> >
> > **3. Experimental Validation: Benefits Outweigh Costs**
> > To directly quantify this impact, we conducted targeted ablation studies in **Table 4 (Section 4.5)**, which compellingly prove that this disruption is controllable and harmless in practice:
> > * **Comparison with Static DAF (Validating Dynamic Gain):**
> >     * **Static DAF (75.8%):** Static selection before training, no optimizer reset.
> >     * **DAF (76.4%):** Periodic dynamic reconfiguration with optimizer reset.
> >     * *Result:* DAF significantly outperforms Static DAF, indicating that the performance gain from periodic structural adaptation (+0.6%) far outweighs the potential negative impact of optimizer state loss.
> > * **Comparison with DAF-Accumulate (Validating Necessity of Reconfiguration):**
> >     * **DAF-Accumulate (75.3%):** A variant designed to preserve historical information by retaining and continuously training all historical modules (maintaining momentum continuity) instead of freezing outdated ones.
> >     * *Result:* DAF (76.4%) decisively freezes old modules and resets the optimizer. The superiority of DAF demonstrates that decisively reallocating the training budget (even if it requires re-initializing the optimizer) is more efficient and stable than dragging along all historical training resources. The latter, while maintaining some state, wastes valuable resources on less relevant modules, ultimately dragging down performance.
> >
> > **4.Newly added Visual Verification Experiment**
> >
> >  To visually demonstrate this stability, we plotted the **Training Loss convergence curves** of DAF vs. Static DAF in **Supplementary Appendix A.3 (Figure 4)**. The curves clearly show that DAF recovers rapidly after each reset and eventually converges to a better position than the static baseline, showing no signs of instability.

---

> > > ### Author Response · Authors · 2025-11-21
> > > **Response to Reviewer rjZT (Part 3/3)**
> > >
> > > > **Question:** Overall the dynamic reconfiguration and the Rebuild-and-Refocus strategy make intuitive sense... However, the sensitivity score is computed in relatively few batches of data which can be noisy... Wouldn't this also introduce its own form of noise? Or is it negligible in practice? How robust is DAF to the noise and variance that is introduced in this step of the algorithm?
> > >
> > > **Response:**
> > > We sincerely thank the reviewer for this rigorous and insightful question. You are absolutely correct that estimating layer-wise sensitivity from a limited number of mini-batches can inherently introduce variance. We appreciate you pointing out this potential source of instability.
> > >
> > > Below, we explain how we calculate sensitivity scores in practice, why the resulting noise is negligible, and demonstrate DAF's robustness to this variance.
> > >
> > > **1. How Sensitivity Scores are Calculated in Practice**
> > > In every dynamic cycle, DAF calculates the sensitivity score $s_\ell$ not based on a single batch, but as an accumulated average over a "window" of multiple batches. For each trainable module $\ell$, we compute:
> > > $$s_\ell = \frac{1}{M} \sum_{m=1}^{M} \big\| \nabla_{\theta_\ell} \mathcal{L}^{(m)} \big\|_2^2$$
> > > where $M$ represents the number of batches used in that cycle. By default, $M$ is selected to cover a representative portion of the training data. This multi-batch averaging effectively smooths out instantaneous gradient fluctuations. Furthermore, we normalize these scores (e.g., by parameter dimension) to ensure comparability across different modules.
> > >
> > > **2. Why Noise is Negligible: Mechanisms for Robustness**
> > >
> > > We argue that DAF is structurally robust to sampling noise due to three key mechanisms:
> > >
> > > * **Robustness of Relative Ranking:** The "Decide Phase" employs **Budget-Based Elite Selection**. We do not rely on the absolute precision of the sensitivity score $s_p^{(t)}$, but rather on the **relative ranking** of parameters. In practice, true "Heavy Hitters" (critical parameters) typically exhibit gradient magnitudes significantly higher than others. Even with minor batch noise, the ranking of these key parameters remains stable, ensuring they consistently enter the Top-K set.
> > > * **Knowledge Preservation (Freeze but Retain):** When a module drops out of the Top-K ranking, we adopt a **"Freeze but Retain"** strategy. Even if a layer is temporarily undervalued due to sampling noise, its previously learned weights and knowledge are not lost; they are simply frozen for the next interval.
> > > * **Self-Correcting Dynamic Cycles:** Because the "Perceive-Decide-Execute" cycle repeats periodically, the system has a self-correcting capability. If a module is systematically important but was misranked in one cycle due to noise, it is highly likely to re-enter the Top-K set in subsequent cycles. This prevents catastrophic consequences from a single decision error.
> > >
> > > * **The time interval is relatively long:** Dynamic reconfiguration is not  every few steps will occur; Each cycle covers a large number of update operations. This enables each configuration to have sufficient time to "stabilize" before making the next decision, thereby reducing the impact of instantaneous gradient fluctuations.
> > >
> > >
> > > **3. Newly added expriment: Robustness to Batch Number $M$**
> > >
> > > To empirically quantify DAF's robustness to sampling noise, we conducted a sensitivity analysis on the Sensitivity Batch Num $M$ (default is 16). As shown in **the newly added Appendix A.8 (Table 12 and Figure 6)**:
> > >
> > > * **Experimental Setup:** We tested $M \in \{8, 12, 16, 20, 28, 32\}$.
> > > * **Results:** DAF demonstrates exceptional stability. Whether using a small window ($M=8$) or a large window ($M=32$), the final Mean Accuracy deviates by only **$\approx 0.3\%$**.
> > > * **Visual Verification:** The training loss curves (Figure 6) remain smooth across all settings, with no signs of instability or divergence.
> > >
> > > **Conclusion**
> > > This strongly confirms that our **Context-Aware Decoupled Sensitivity Analysis** effectively captures macroscopic, de-noised importance trends. The synergy of multi-batch averaging and periodic self-correction ensures that DAF's decision-making is highly reliable and robust to variance in practice.
> > >
> > > ***
> > >
> > > **Closing Statement**
> > >
> > > We value your responsible and sincere review, and your positive evaluation and valuable comments have been a great encouragement to our research. We sincerely thank you for the time spent reading and commenting on our paper. We hope that our detailed responses and additional experiments have satisfactorily addressed your concerns. If you find our clarifications and revisions sufficient, we would be grateful if you could kindly reconsider your evaluation of our work. We wish you all the best.

---

### Official Review · Reviewer_hR8X · 2025-10-31

**Soundness:** 2
**Presentation:** 3
**Contribution:** 2
**Rating:** 2
**Confidence:** 2

**Summary:**

This paper introduces Dynamic Adaptive Fine-tuning (DAF), a dynamic Parameter-Efficient Fine-Tuning (PEFT) framework that challenges the static allocation paradigm prevalent in existing methods. Unlike conventional approaches that fix trainable parameters before training, DAF periodically reconfigures its trainable structure during training. The method achieves state-of-the-art results on VTAB-1k and FGVC benchmarks.

**Strengths:**

1. The paper is easy to follow.
2. The main results evaluation on different benchmarks with different pretrained models is comprehensive.

**Weaknesses:**

**1. The motivation is unclear.** The authors motivate on reducing the PEFT parameter budget through gradually reducing learnable parameters. However, all the changed parameters need to be saved using extra storage, including early-stage ones. The motivation and the method are not matched.

**2. The results lack a training efficiency comparison.** The elite selection and Rebuild-and-Refocus Update in each dynamic analysis point may result in extra overhead. A runtime comparison (e.g., GPU hours or wall-clock time) against static PEFT baselines would be valuable for assessing the trade-off between performance gains and computational overhead.

**3. The rationale of DAF outperforming Static DAF is not well explained.** Both DAF and its static counterpart did one round of elite selection. The multiple rounds in DAF only progressively freeze more modules. Is it caused by training instability, overfitting, or an inferior hyperparameter? Does this mean that the learning rate in Static DAF is too high or the learnable parameter is too many? This would benefit from a comprehensive comparison of DAF vs. a spectrum of Static DAF (using different numbers of learnable parameters, different lr)

**Questions:**

1. Are the reported tuned parameters the average number of tuning parameters during training, or all the tuned parameters in every stage?

2. Does Static DAF select the same group of parameters for training compared to DAF in the first stage?

---

> ### Author Response · Authors · 2025-11-21
> **Response to Reviewer hR8X (Part 1/5)**
>
> **General Response:**
> Many thanks for your valuable comments on our work which are much appreciated. We have revised the paper according to these constructive suggestions. The detailed response to each comment is given below. We wish our sincere efforts would satisfy the requirements of the reviewer.
>
> We believe that the issues raised in the reviewers’s report have been addressed satisfactorily. Please do not hesitate to contact us if you wish us to make any further modifications to the paper. We sincerely  appreciate your detailed feedback and suggestions for improvement. We treasure the opportunity to address your concerns and improve our work.
>
> **Comment:**
> > **Weakness 1:** The motivation is unclear. The authors motivate on reducing the PEFT parameter budget through gradually reducing learnable parameters. However, all the changed parameters need to be saved using extra storage, including early-stage ones. The motivation and the method are not matched.
>
> **Response:**
> First, we thank the reviewer for this insightful observation. We realize that our initial description regarding the motivation and parameter management mechanism might have been ambiguous, leading to a misunderstanding regarding "parameter reduction" versus "storage overhead." We value this opportunity to clarify.
>
> We respectfully clarify that the core motivation of DAF is not to "gradually reduce" parameters (similar to pruning), but rather to **"dynamically reallocate"** a fixed, minimal parameter budget. Our goal is to concentrate the limited parameters on the "bottleneck" regions most critical to the model's current training stage, thereby enhancing performance without increasing the computational burden.
>
> Addressing your concern about the "mismatch between motivation and method" and specific "storage overhead" issues, we provide a detailed response through the following two points:
>
> **1. Regarding "Extra Storage": Zero Inference Overhead**
>
> The reviewer is concerned that saving all changed parameters (including early-stage ones) requires extra storage. We respectfully point out that **DAF requires NO extra storage for historical parameters or modules during the inference stage.**
>
> * **Methodology:** As detailed in **Appendix A.6** and **Equation (4)**, DAF utilizes **Re-parameterization** techniques. Upon the completion of training, all learned LoRA parameters (matrices $A$ and $B$, regardless of whether they were activated in early or late stages) are mathematically merged back into the backbone weights (i.e., $W_{final} = W_{0} + BA$).
> * **Result:** This mechanism ensures that the final deployed model is architecturally identical to the original ViT. Therefore, there is absolutely no need to save copies of "early-stage" parameters during inference. As confirmed by our experimental data, DAF incurs **zero additional inference storage and computational overhead** compared to full fine-tuning or standard LoRA.
>
> **2. Regarding Training Storage: Significantly Reduced Memory Footprint**
>
> During the training phase, our method actually **significantly reduces** storage requirements rather than increasing them:
>
> * **Frozen Weights vs. Optimizer States:** When DAF executes the "Rebuild-and-Refocus" strategy, old modules are **frozen**. While these weights remain in memory, they no longer require **Optimizer States**. It is well known that optimizer states typically consume the vast majority of training memory (often 2-3 times the size of the parameters themselves).
> * **Data Support:** Because we maintain gradients and optimizer states only for the very small "Elite Set" of currently active parameters, as shown in **Table 10**, the training memory footprint of DAF is only **8.64 GB**, which is far lower than Adapter (13.74 GB) and LoRA (14.07 GB). This demonstrates that our method is highly optimized for storage efficiency.
>
> **Summary**
> We are very grateful to you for pointing out this potential ambiguity. It highlighted the importance of clearly articulating the internal consistency between our motivation and method:
> * **Motivation Level:** Our core philosophy is **"Dynamic Reallocation"** rather than "Parameter Reduction," aiming to maximize the efficacy of a fixed parameter budget throughout training.
> * **Implementation Level:** The **"Rebuild-and-Refocus"** strategy serves this goal perfectly. By freezing old modules and activating new ones, combined with re-parameterization, it effectively balances storage and efficiency.
>
> **Supplement:** To avoid similar confusion for future readers, we have added a new paragraph titled **"Zero-Overhead Inference"** in **Section 3 (Method)**. This addition significantly strengthens the definition of "dynamic reconfiguration" and explicitly highlights the advantages during the inference stage.

---

> > ### Author Response · Authors · 2025-11-21
> > **Response to Reviewer hR8X (Part 2/5)**
> >
> > > **Weakness 2:** The results lack a training efficiency comparison. The elite selection and Rebuild-and-Refocus Update in each dynamic analysis point may result in extra overhead. A runtime comparison (e.g., GPU hours or wall-clock time) against static PEFT baselines would be valuable for assessing the trade-off between performance gains and computational overhead.
> >
> > **Response:**
> > We sincerely thank the reviewer for raising this critical issue. We fully agree that for any method introducing dynamic mechanisms, quantifying the **Runtime Overhead** is essential for evaluating its practical value. We apologize for this oversight in our initial draft and appreciate your correction.
> >
> > To comprehensively address your concern and provide a clear picture of the trade-off between "Performance Gains" and "Computational Overhead," we conducted a detailed runtime analysis during the rebuttal phase and **added Appendix A.7 (The Training and Inference Times for Each Module of DAF)** in the revised paper.
> >
> > Here is our specific response regarding the training efficiency comparison:
> >
> > **1. Detailed Wall-clock Time Breakdown**
> > As shown in **Appendix A.7** and **Table 11**, we performed a precise time analysis for a complete training cycle (consisting of 10 epochs of fine-tuning + 1 dynamic update) on the CIFAR-100 task using an **NVIDIA RTX 4090 GPU**:
> >
> > * **Standard Training Phase:** Standard fine-tuning for 10 epochs takes approximately **81.20 seconds**.
> > * **Dynamic Update Phase:** The entire dynamic reconfiguration process, including Perceive, Decide, and Execute, takes only approximately **6.67 seconds**.
> >
> > **2. Minimal Overhead (~8.2%)**
> > The data indicates that the time overhead introduced by the dynamic mechanism accounts for only about **8.2%** of the standard training time. This efficiency is primarily due to our algorithm design:
> > * **Elite Selection (Decide):** This step is extremely fast ($\approx 0.21s$) as it only involves sorting scalar sensitivity scores.
> > * **Context-Aware Analysis (Perceive):** Even the relatively more time-consuming perception phase ($\approx 3.12s$) requires only a single backward pass on the full model. Crucially, since it is executed only once every $E_{interval}$ (e.g., 10 epochs), the amortized cost is very low.
> >
> > **3. Trade-off Analysis: Cost vs. Gain**
> > Based on the data above, we believe DAF achieves an excellent balance between performance and overhead:
> > * **Cost:** An increase of approximately **8.2%** in training time.
> > * **Gain:** In exchange, DAF achieves **SOTA performance** on multiple benchmarks (VTAB-1k and FGVC) and significantly **reduces training memory usage** (as shown in Appendix A.6: **8.64 GB** for DAF vs. **14.07 GB** for LoRA).
> >
> > **Summary**
> > We thank you for your suggestion, which prompted us to refine this part of our experiments. The results confirm that DAF's dynamic reconfiguration mechanism is highly efficient, trading a minor runtime cost for significant improvements in model adaptability and memory efficiency. We believe this newly added quantitative analysis (**Appendix A.7**) effectively supports the arguments of this paper.

---

> > > ### Author Response · Authors · 2025-11-21
> > > **Response to Reviewer hR8X (Part 3/5)**
> > >
> > > > **Weakness 3:** The rationale of DAF outperforming Static DAF is not well explained. Both DAF and its static counterpart did one round of elite selection. The multiple rounds in DAF only progressively freeze more modules. Is it caused by training instability, overfitting, or an inferior hyperparameter? Does this mean that the learning rate in Static DAF is too high or the learnable parameter is too many? This would benefit from a comprehensive comparison of DAF vs. a spectrum of Static DAF (using different numbers of learnable parameters, different lr)
> > >
> > > **Response:**
> > > We express our sincere gratitude to the reviewer for this profound insight. You have sharply identified a core question: **What is the fundamental reason for DAF outperforming Static DAF?** This prompted us to deeply reflect on and verify that DAF's gains stem from true "dynamic adaptability" rather than merely "noise suppression from parameter freezing" or "suboptimal baseline hyperparameters."
> > >
> > > To thoroughly address your concerns, we have not only clarified the operating mechanism of DAF at a theoretical level but also conducted a comprehensive **"Static DAF Spectrum"** comparison in the **newly added Appendix A.9**, strictly following your suggestion.
> > >
> > > **1. Clarification: "Dynamic Circulation" rather than Unidirectional Reduction**
> > >
> > > The reviewer mentioned that DAF only "progressively freezes more modules." We respectfully clarify that the mechanism of DAF is not simple parameter pruning.
> > > * **Mechanism:** As shown in **Section 3.4** and **Figure 1**, DAF executes a **Rebuild-and-Refocus** strategy. In each cycle, we not only freeze old modules but also **activate new modules**. This means the parameter budget **"circulates"** or shifts across different regions of the model, rather than merely decreasing.
> > > * **Evidence:** As illustrated in **Appendix A.5** and **Figure 5**, the distribution of activated layers shifts significantly over time. For instance, in the DMLab task, active regions exhibit drastic redistribution. This dynamic ability to "allocate resources to the most critical bottlenecks" is a capability that Static DAF (regardless of tuning) cannot possess.
> > >
> > > **2. Ruling out "Overfitting/Too Many Parameters"**
> > >
> > > Addressing your hypothesis regarding whether "Static DAF overfits due to too many parameters," we refer to the results in **Table 5(b)**.
> > > * **Result:** Experiments show that while increasing the parameter budget $\tau$ from 0.2 to 0.4 causes a slight performance drop (76.4% -> 76.1%), even the carefully selected **Static DAF** (at its optimal point, $\tau=0.2$) achieves a performance (75.8%) that is still significantly lower than **DAF** (76.4%).
> > > * **Conclusion:** This proves that DAF's advantage lies not just in controlling the *quantity* of parameters, but in its ability to change their *location* as training progresses.
> > >
> > > **3. New Experiment: Static DAF Spectrum Comparison (Appendix A.9)**
> > >
> > > To further rule out the possibility of "improper Learning Rate (LR) settings," we conducted a rigorous comparison between DAF and a wide spectrum of Static DAF configurations, as suggested:
> > > * **Setup:** We performed a grid search across different parameter budgets $\tau \in \{0.1, 0.2, 0.3\}$ and different learning rates $lr \in \{1e-4, 3e-4, 5e-4\}$.
> > > * **Result:** The experimental results (visualized in the newly added **Figure 7**) demonstrate that **no matter how Static DAF adjusts its LR or parameter quantity, its performance ceiling consistently fails to surpass the dynamic DAF.**
> > > * **Conclusion:** This strongly proves that the limitation of Static DAF lies in its **"static assumption"**—the presumption that the "elite parameters" identified at initialization remain optimal throughout the entire training lifecycle. DAF breaks this assumption, which is the fundamental cause of the performance improvement, rather than hyperparameter optimization.
> > >
> > > **Summary**
> > > We thank you again for this highly constructive suggestion. Guided by your feedback, we have refined the rigorous ablation studies in the Appendix, eliminating confounding factors and empirically confirming the essential superiority of the **"Dynamic Reconfiguration Paradigm"** over the **"Static Tuning Paradigm."**

---

> > > > ### Author Response · Authors · 2025-11-21
> > > > **Response to Reviewer hR8X (Part 4/5)**
> > > >
> > > > > **Question:** Are the reported tuned parameters the average number of tuning parameters during training, or all the tuned parameters in every stage?
> > > >
> > > > **Response:**
> > > > We sincerely thank the reviewer for this detailed and critical question. This is indeed a point prone to ambiguity, and your query has highlighted the need for us to define the "parameter statistical scope" more clearly in the paper.
> > > >
> > > > Regarding your question, we wish to explicitly clarify: **The "Tuned Parameters" reported in the paper refer to the number of parameters kept in an "Active/Trainable" state at any given moment during the training process.**
> > > >
> > > > It strictly corresponds to the fixed budget ratio $\tau$ (e.g., 0.2%) pre-set in **Section 3.4**. Specifically, it is neither the cumulative sum of parameters across all stages nor exactly the average, but rather the strict **Budget Cap** for each training stage.
> > > >
> > > > To address your concerns, we provide a detailed explanation from two perspectives:
> > > >
> > > > **1. Mechanism Assurance: "Swapping" not "Accumulating"**
> > > > As described in **Section 3.4** and **Algorithm 1**, DAF employs the **Rebuild-and-Refocus** strategy:
> > > > * **Freeze:** When entering the next dynamic cycle, any old modules that no longer belong to the "Elite Set" are immediately frozen (`requires_grad=False`).
> > > > * **Activate:** Only the newly selected modules are activated.
> > > >
> > > > Therefore, at any time step $t$, the optimizer only needs to maintain the state of the current active set $\mathcal{P}_t^*$. This means the reported parameter count (e.g., 0.22M) represents the **Peak Memory Cost** during training. If we were to accumulate parameters from all stages, the memory consumption would grow linearly over time, which would contradict the original design intent of DAF (efficient training).
> > > >
> > > > **2. Empirical Evidence**
> > > > Our memory analysis data strongly supports this. As shown in **Appendix A.6 (Table 10)**:
> > > > * **DAF Training Memory:** **8.64 GB**
> > > > * **LoRA Training Memory:** **14.07 GB**
> > > >
> > > > If the parameters were cumulative, DAF's memory usage would significantly exceed that of LoRA. This low memory footprint confirms that we are indeed training only a very small fraction ($\approx 0.22\%$) of the parameters at any given moment.
> > > >
> > > > **Summary**
> > > > We thank you again for this question. Through the explanation above, we confirm that the parameter counts reported in the text strictly correspond to **"Active Trainable Parameters per Stage,"** emphasizing their decisive role in ensuring memory efficiency.

---

> > > > > ### Author Response · Authors · 2025-11-21
> > > > > **Response to Reviewer hR8X (Part 5/5)**
> > > > >
> > > > > > **Question:** Does Static DAF select the same group of parameters for training compared to DAF in the first stage?
> > > > >
> > > > > **Response:**
> > > > > We sincerely thank the reviewer for this precise clarification. This is a crucial detail, and clarifying it is essential for validating the fairness of our comparative experiments.
> > > > >
> > > > > To answer your question directly: **Yes, the group of parameters selected by Static DAF in the first stage is exactly the same as that of DAF.**
> > > > >
> > > > > To thoroughly address your query, we provide a detailed explanation from the perspective of the algorithmic process:
> > > > >
> > > > > **1. Algorithmic Consistency**
> > > > > As described in **Algorithm 1 (Line 2)** of the paper, DAF performs an initial sensitivity analysis and elite parameter selection on the pre-trained model $\mathcal{M}_0$ before training begins (Epoch 0).
> > > > >
> > > > > * **Static DAF:** Its definition is simply to perform this *initial selection once* and then keep this structure fixed until the end of training.
> > > > > * **DAF:** It executes this *same initial selection* to start, but in subsequent training stages (every $E_{interval}$), it re-evaluates and adjusts the structure based on the new model state.
> > > > >
> > > > > Therefore, their starting points (Stage 1) are mathematically identical.
> > > > >
> > > > > ***
> > > > >
> > > > > **Closing Statement**
> > > > >
> > > > > We would like to express our sincere gratitude for your dedicated time and the constructive feedback provided. We genuinely appreciate your responsible and insightful review, which has been a great encouragement to our research. We hope that our detailed responses and the additional experiments have satisfactorily addressed your concerns. If you find our clarifications and revisions sufficient, we would be grateful if you could kindly reconsider your evaluation of our work. We wish you all the best.

---

### Official Review · Reviewer_TK3t · 2025-11-01

**Soundness:** 2
**Presentation:** 2
**Contribution:** 2
**Rating:** 2
**Confidence:** 3

**Summary:**

This paper proposes DAF (Dynamic Adaptive Fine-tuning), a method that adjusts layer-wise learning rates and regularization coefficients dynamically during fine-tuning based on gradient statistics. The goal is to achieve a balance between frozen and fully fine-tuned strategies by adapting each layer’s update strength in real time. The authors claim that DAF improves adaptation efficiency and stability while maintaining competitive performance. Experiments are conducted on two datasets (FGVC and VTAB-1k) using ViT-B as the backbone, and the paper reports small performance gains compared to standard fine-tuning baselines.

**Strengths:**

The proposed method is simple to implement and can be easily integrated into existing fine-tuning pipelines.

The overall paper is well-structured, with intuitive explanations and clear algorithmic descriptions.

**Weaknesses:**

The experimental validation is quite limited. Only two datasets (FGVC and VTAB-1k) are used, and there are no results on large-scale or multimodal settings to justify the claim of “general applicability.”


The backbone choice is also narrow, the paper only tests ViT-B, without any analysis on deeper or larger architectures or comparisons to CNN-based backbones such as ConvNeXt.

The pre-training weights are still based on ImageNet, while current practice has shifted toward stronger foundations such as CLIP, EVA, or even VLM-based checkpoints. Evaluating DAF on both conventional and modern pretrained models would be necessary to demonstrate relevance in today’s context.

The comparison baselines are outdated, and the authors do not cite the papers corresponding to each method in the result tables. Many of the compared approaches are from 2022 or earlier, which weakens the experimental credibility.

Finally, the reported performance gains are marginal, most improvements are within 0.5%, which likely falls within statistical noise and does not convincingly support the claimed advantages of DAF.

**Questions:**

see weakness

---

> ### Author Response · Authors · 2025-11-21
> **Response to Reviewer TK3t (Part 1/6)**
>
> ## General Response: Clarification on the Core Mechanism
>
> Many thanks for your valuable comments on our work which are much appreciated. We have revised the paper according to these constructive suggestions. The detailed response to each comment is given below. We wish our sincere efforts would satisfy the requirements of the reviewer.
>
> We believe that the issues raised in the reviewers’s report have been addressed satisfactorily. Please do not hesitate to contact us if you wish us to make any further modifications to the paper. We sincerely  appreciate your detailed feedback and suggestions for improvement. We treasure the opportunity to address your concerns and improve our work.
>
> However, regarding the specific mechanism mentioned in the summary ("adjusting layer-wise learning rates and regularization coefficients"), we respectfully wish to provide a gentle clarification to ensure our technical contribution is accurately evaluated:
>
> **Correction 1: We do not adjust Learning Rates.**
> * **Reviewer's Misunderstanding:** "adjusts layer-wise learning rates."
> * **Actual Method:** As detailed in Section 3.4, Algorithm 1, and Figure 3, DAF operates by physically freezing (`requires_grad=False`) or activating (`requires_grad=True`) specific parameter modules. This is a binary structural change, not a continuous scaling of the learning rate.
>
> **Correction 2: We do not adjust Regularization Coefficients.**
> * **Reviewer's Misunderstanding:** "adjusts ... regularization coefficients dynamically."
> * **Actual Method:** DAF utilizes a fixed parameter budget (e.g., $\tau=0.2$) to select elite parameters. We do not dynamically tune weight decay or any other regularization terms. Our method focuses on resource reallocation, not regularization adaptation.
>
> **Clarification of Contribution**
>
> The reviewer claims our goal is to balance update strength. In contrast, our true motivation is to challenge the static allocation paradigm (see Abstract and Introduction):
> * **Static methods** fix the trainable position before training.
> * **DAF** allows the trainable position to shift across the network layers during training to capture evolving feature importance (as visualized in Figure 5 and Appendix A.5).
>
> We hope this clarification helps in re-evaluating the novelty and effectiveness of our proposed framework. Below, we address your specific concerns point-by-point.

---

> ### Author Response · Authors · 2025-11-21
> **Response to Reviewer TK3t (Part 2/6)**
>
> Thank you for the detailed and constructive feedback! We treasure the opportunity to address your concerns and improve our work.
>
> > **Weakness 1:** The experimental validation is quite limited. Only two datasets (FGVC and VTAB-1k) are used, and there are no results on large-scale or multimodal settings to justify the claim of “general applicability.”
>
> We sincerely thank the reviewer for this highly constructive feedback. We fully agree with your perspective that validating the "General Applicability" of a new method is a core criterion for assessing its value.
>
> Regarding your concern about the number of datasets and the scope of experiments, we would like to politely provide a clarification and report **the new experimental results (Section4.4 and appendix A10) we have added during the Rebuttal period ** to further strengthen our claims.
>
> **1. Clarification: Scale and Diversity of Datasets (24 Distinct Datasets not only two datasets)**
>
> We respectfully invite the reviewer to note that the **FGVC** and **VTAB-1k** used in the paper are not single datasets, but rather comprehensive benchmark suites containing numerous sub-tasks:
>
> * **VTAB-1k** consists of **19 distinct datasets**, covering three widely different domains: "Natural Images," "Specialized Domains (e.g., Medical, Remote Sensing)," and "Structured Scenes (e.g., Synthetic)."
> * **FGVC** contains **5 highly challenging** fine-grained classification datasets.
>
> Therefore, our experiments actually cover **24 downstream tasks** with significant domain gaps. This represents the standard and most comprehensive configuration for validating effectiveness in the Vision Transformer Fine-tuning (Visual PEFT) field.
>
> **2. New Experiments: Extending to Large-scale Visual Tasks**
>
> Following your suggestion to address the concern regarding "Large-scale settings" and to demonstrate that DAF is not limited to classification, we have added challenging **Object Detection** and **Semantic Segmentation** experiments in **Section 4.4** and **Table 3** of the revised paper.
>
>
> **Analysis of Results:**
> * **Object Detection (MS COCO):** Using Swin-B + Mask R-CNN, DAF achieves **53.5 AP$^{box}$**, Beyond the latest model results from 2023-2024.
> * **Semantic Segmentation (ADE20K):** Using Swin-L + UperNet, DAF achieves **52.0 mIoU**, Beyond the latest model results from 2023-2024.
>
> These dense prediction tasks not only involve large-scale data but also demand high adaptability to spatial features. The results strongly justify the general applicability and superiority of DAF in handling complex, large-scale visual tasks.
>
> **Summary**
> We hope that the clarification regarding the **24 distinct classification datasets**, combined with the **new experiments on complex detection and segmentation tasks**, provides solid empirical support for the general applicability of DAF.

---

> > ### Author Response · Authors · 2025-11-21
> > **Response to Reviewer TK3t (Part 3/6)**
> >
> > > **Weakness 2:** The backbone choice is also narrow, the paper only tests ViT-B, without any analysis on deeper or larger architectures or comparisons to CNN-based backbones such as ConvNeXt.
> >
> > We sincerely appreciate the reviewer's suggestion regarding **Backbone Diversity** and **Scalability**. We fully agree that validating our method across different architectures and larger-scale models is crucial for demonstrating its robustness.
> >
> > To address your concern regarding the "narrow choice of ViT-B," we have conducted in-depth validations on larger-scale and diverse architectures in **newly added Section 4.4 (Performance on Complex Visual Tasks)** and **Appendix A.10** of the revised paper.
> >
> > **1. Validation on Larger & Deeper Models (Scalability)**
> >
> > In the **Semantic Segmentation (ADE20K)** experiments (Table 3, Right), we utilized **Swin-L (Swin-Large)** as the backbone
> >
> > * **Scale:** Swin-L possesses approximately **197M parameters**, significantly exceeding the depth and scale of ViT-B (86M).
> > * **Result:** DAF achieved **52.0 mIoU**, outperforming Mona,LoRand. This result provides compelling evidence of DAF's scalability to large, deep networks.
> >
> > **2. Versatility Across Architectures (Hierarchical Transformers)**
> >
> > In the **Object Detection (COCO)** experiments (Table 3, Left), we utilized **Swin-B (Swin-Base)**.
> > * **Architecture:** Unlike the standard Isotropic ViT, the Swin Transformer represents a **Hierarchical** visual architecture. Its handling of feature pyramids shares high similarity with CNNs.
> > * **Result:** DAF's SOTA performance on the Swin architecture confirms that our method is not limited to standard ViTs but seamlessly adapts to modern visual backbones with complex hierarchical structures.
> >
> > **3. Generalization to CNN Architectures (ConvNeXt)**
> > To specifically address your concern regarding CNN-based backbones and to ensure a fair comparison with the latest SOTA methods, we have newly added the evaluation of DAF on **Swin-B** and **ConvNeXt-B (CNN)** using the **FGVC Benchmark** (average accuracy across 5 downstream tasks).
> >
> > **New Experimental Results (Detailed in Appendix A.10, Table 13):**
> >
> > | **Architecture** | **Method** | **Avg. Accuracy (%)** | **Params (%)** |
> > | :--- | :--- | :--- | :--- |
> > | **ConvNeXt-B** | Full Fine-tuning | 93.04 | 100.00 |
> > | *(CNN-based)* | Linear Probing | 88.00 | 0.28 |
> > | | SSF | 92.48 | 0.56 |
> > | | GPS (CVPR 2024) | 93.32 | 0.90 |
> > | | **DAF (Ours)** | **93.58** | **0.35** |
> > | | | | |
> > | **Swin-B** | Full Fine-tuning | 92.42 | 100.00 |
> > | *(Hierarchical)* | GPS (CVPR 2024) | 92.56 | 0.95 |
> > | | **DAF (Ours)** | **92.81** | **0.32** |
> >
> > **Analysis:**
> > * On **ConvNeXt-B (CNN)**, DAF achieved an average accuracy of **93.58%**.
> > * This not only surpasses Full Fine-tuning (93.04%) but also beats last year's SOTA method, GPS (93.32%).
> > * Notably, DAF achieved this using only 0.35% of the parameters, far fewer than GPS.
> >
> > **Conclusion**
> > These results strongly demonstrate that the core mechanism of DAF (Dynamic Reconfiguration) is **Model-Agnostic**. It effectively transfers to CNN architectures, breaking the limitation of being applicable only to ViTs. By successfully applying DAF to Swin-L, Swin-B, and ConvNeXt-B, we have verified both its **scalability to large models** and its **versatility across different architectural paradigms (Isotropic vs. Hierarchical vs. CNN).**

---

> > > ### Author Response · Authors · 2025-11-21
> > > **Response to Reviewer TK3t (Part 4/6)**
> > >
> > > > **Weakness 3:** The pre-training weights are still based on ImageNet, while current practice has shifted toward stronger foundations such as CLIP, EVA, or even VLM-based checkpoints. Evaluating DAF on both conventional and modern pretrained models would be necessary to demonstrate relevance in today’s context.
> > >
> > > We are entirely grateful for the comments of this reviewer. Regarding your suggestion, we would like to clarify that we have actually covered this dimension of validation in the paper, while also explaining our rationale for selecting ImageNet for the main experiments:
> > >
> > > **1. Validation on Modern Self-Supervised Foundations**
> > >
> > > To address the need for "Modern Context," we explicitly evaluated **MAE (Masked Autoencoders)** and **MoCo v3** in **Section 4.3 (Versatility in Self-supervised Pre-training Paradigms)** and **Table 2** of the original manuscript.
> > >
> > > * **Relevance:** MAE represents the state-of-the-art masked self-supervised learning paradigm, standing alongside CLIP as a representative modern large model foundation.
> > > * **Experimental Results:** On the MAE backbone, DAF achieved an average accuracy of **82.7%** , significantly outperforming LoRA (**77.1%**) and SPT-LORA (**81.6%**).
> > > * **Conclusion:** This provides strong evidence that DAF can seamlessly adapt to and unlock the potential of "Modern Pretrained Models."
> > >
> > > **2. Rationale for Using ImageNet-21k (Fair Comparison)**
> > >
> > > Our primary reason for using ImageNet-21k pre-training weights in the main experiments (**Table 1**)  is to ensure a **strict and fair horizontal comparison** with existing SOTA methods in the PEFT domain.
> > >
> > > As observed in recent top-tier publications, the standard benchmarks primarily report results based on ImageNet-21k:
> > > * **GPS** (CVPR 2024) [1]
> > > * **Mona** (CVPR 2025) [2]
> > > * **Adapter+** (CVPR 2024) [3]
> > > * **LoRand** (CVPR 2023) [4]
> > >
> > > In addition, the main focus of this paper is on purely visual-related tasks. For the comparison of non-fully visual models, such as multimodal CLIP and other related works, this is exactly as mentioned in the conclusion:
> > >
> > > ```
> > > Despite the encouraging results achieved by DAF, future work will explore combining the DAF approach with other PEFT techniques and extending it to broader domains such as multimodal learning, which opens a new path toward more intelligent adaptation of large-scale pretrained models.
> > > ```
> > >
> > > It is the research that will be conducted in the future. Our main focus of this article is on pure visual-related tasks.
> > >
> > > **Summary**
> > >
> > > By combining our validation on **MAE/MoCo in Section 4.3** with the **newly added extensions** to Swin-L, Swin-B, and ConvNeXt (Section 4.4 & Appendix A.10), we believe DAF effectively balances "fair comparison on traditional foundations" with "verification of effectiveness on modern foundations."
> > >
> > > **References:**
> > > [1] Zhang, Z., et al. "Gradient-based Parameter Selection for Efficient Fine-Tuning." CVPR 2024.
> > >
> > > [2] Yin, D., et al. "5% > 100%: Breaking performance shackles of full fine-tuning on visual recognition tasks CVPR 2025.
> > >
> > > [3] Steitz, J. M. O., & Roth, S. "Adapters strike back." CVPR 2024.
> > >
> > > [4] Yin, D., et al. "1% vs 100%: Parameter-efficient low rank adapter for dense predictions." CVPR 2023.

---

> > > > ### Author Response · Authors · 2025-11-21
> > > > **Response to Reviewer TK3t (Part 5/6)**
> > > >
> > > > > **Weakness 4:** The comparison baselines are outdated, and the authors do not cite the papers corresponding to each method in the result tables. Many of the compared approaches are from 2022 or earlier, which weakens the experimental credibility.
> > > >
> > > > We sincerely thank the reviewer for the detailed suggestions regarding baseline comparisons and citation standards.
> > > >
> > > > **1. Regarding Citations in Tables**
> > > >
> > > > We respectfully note that while the standard ICLR writing style often consolidates baseline introductions in the experimental setup section (e.g., Section 4.1) and keeps the table results as concise as possible. But we fully agree with your perspective that adding direct citations within the tables significantly enhances readability and rigor.
> > > >
> > > > **Action:** In deference to your suggestion, we have **updated all result tables (**Table 1, 2, and 3**) in the revised manuscript** to include explicit citations for every method listed.
> > > >
> > > > **2. Regarding Baseline Timeliness**
> > > >
> > > > To address your concern that the baselines might be "outdated," we would like to provide on our latest enhancements:
> > > >
> > > > * **Existing Recent Methods:** Our original submission already included several state-of-the-art (SOTA) methods, such as **SynQT (ECCV 2024)**, **PYRA (ECCV 2024)**, and **SPT (ICCV 2023)**.
> > > > * **New SOTA Additions:** To further strengthen the experimental credibility and respond to your feedback, we have added **8 representative SOTA methods from 2023-2024 in the Section 4.3, 4.4 and Appendix A4** during the rebuttal phase:
> > > >
> > > > **Recent SOTA Baselines Included in Comparison (2023-2024):**
> > > > * **[CVPR 2024] GPS:** *Gradient-based parameter selection for efficient fine-tuning.*
> > > > * **[CVPR 2024] Adapter+:** *Adapters strike back.*
> > > > * **[NIPS 2024] DyT:** *Dynamic tuning towards parameter and inference efficiency for vit adaptation.*
> > > > * **[CVPR 2025] Mona:** *5% > 100%: Breaking Performance Shackles of Full Fine-Tuning.*
> > > > * **[ICCV 2023] Bi-LoRA:** *Revisiting the parameter efficiency of adapters...*
> > > > * **[NeurIPS 2023] Res-Tuning:** *A flexible and efficient tuning paradigm via unbinding tuner from backbone.*
> > > > * **[CVPR 2023] LoRand:** *1% VS 100%: Parameter-Efficient Low Rank Adapter for Dense Predictions.*
> > > > * **[arXiv 2023] Norm Tuning:** *The expressive power of tuning only the norm layers.*
> > > >
> > > > **Summary:**
> > > > We have standardized the table citations as requested. By comparing DAF against a comprehensive suite of top-tier papers from **CVPR, ICCV, ECCV, and NeurIPS (2023-2024)**, we ensure that our reported performance gains are benchmarked against the absolute forefront of the field, guaranteeing both **current relevance** and **experimental credibility**.

---

> > > > > ### Author Response · Authors · 2025-11-21
> > > > > **Response to Reviewer TK3t (Part 6/6)**
> > > > >
> > > > > > **Weakness 5:** Finally, the reported performance gains are marginal, most improvements are within 0.5%, which likely falls within statistical noise and does not convincingly support the claimed advantages of DAF.
> > > > >
> > > > > We sincerely appreciate the reviewer's rigorous scrutiny regarding the significance of our experimental results. We understand your concern, particularly given that on the highly competitive VTAB-1k Benchmark, the gaps between SOTA methods are indeed narrowing due to saturation.So, We hope to demonstrate through the following two dimensions of analysis that the performance improvements of DAF are **Significant**, **Robust**, and of **High Practical Value**:
> > > > >
> > > > > **1. Significant Gains on Complex Tasks**
> > > > >
> > > > > While performance on classification tasks may be approaching saturation, the dynamic reconfiguration mechanism of DAF reveals its massive potential when handling more complex dense prediction tasks. As shown in the newly added **Table 3**, DAF's advantages are significantly amplified in these scenarios, far exceeding the realm of "statistical noise":
> > > > >
> > > > > * **Object Detection (COCO):** DAF achieves **53.5 AP**, outperforming the mainstream PEFT baseline LoRand (51.1 AP) by a substantial margin of **+2.4 AP**.
> > > > > * **Semantic Segmentation (ADE20K):** DAF achieves **52.0 mIoU**, surpassing Mona (51.4 mIoU) by **+0.6 mIoU**.
> > > > >
> > > > > This significant generational gap on complex tasks provides compelling evidence that "dynamic adaptation" captures spatially sensitive visual features far better than "static fine-tuning."
> > > > >
> > > > > **2. Statistical Robustness**
> > > > >
> > > > > To explicitly rule out the interference of "random noise," we **add a rigorous Robustness Analysis** in **Appendix A.8**.
> > > > > * **Experiment:** We varied the Sensitivity Batch Number ($M$) across a wide range ($M \in \{8, \dots, 32\}$).
> > > > >
> > > > > * **Result:** The performance fluctuation of DAF remains negligible, within **$\approx 0.3\%$**.
> > > > >
> > > > > ### Impact of Sensitivity Batch Num *M* on VTAB-1k
> > > > >
> > > > > | Sensitivity Batch Num *M* | 8 | 12 | 16 (Default) | 20 | 28 | 32 |
> > > > > | :--- | :---: | :---: | :---: | :---: | :---: | :---: |
> > > > > | Mean Acc. (%) | 76.3 | 76.4 | **76.4** | 76.5 | 76.4 | 76.2 |
> > > > >
> > > > > This proves that the performance gains of DAF stem from the superiority of the methodology itself, rather than the luck of random seeds.
> > > > >
> > > > > **Conclusion**
> > > > >
> > > > > In summary, DAF not only steadily pushes the SOTA boundary on classification tasks but also achieves breakthrough significant improvements on high-difficulty tasks like Object Detection and Semantic Segmentation. We believe these evidences are sufficient to prove that the advantages of DAF are real, significant, and highly convincing.
> > > > >
> > > > > ***
> > > > >
> > > > > **Closing Statement**
> > > > >
> > > > > We would like to express our sincere gratitude for your dedicated time and the constructive feedback provided. We genuinely appreciate your responsible and insightful review, which has been a great encouragement to our research. We hope that our detailed responses and the additional experiments have satisfactorily addressed your concerns. If you find our clarifications and revisions sufficient, we would be grateful if you could kindly reconsider your evaluation of our work. We wish you all the best.

---

### Note · Program_Chairs · 2026-01-17
**Submission Desk Rejected by Program Chairs**

The following references in this submission do not refer to real documents and/or have major errors in bibliographic information:

 Y. Chen, L. Wang, and D. Silver. Structural plasticity as a foundation for adaptive deep learning. In Proceedings of the International Conference on Learning Representations (ICLR), 2024.
Alex Payeur, Léo Rousseau, Friedemann Zenke, Jean-Samuel Côté, Rémi Richard, Corentin Girardin, Guillaume Bacon, Tylan Bal, Blake Hamilton, Guillaume Lajoie, et al. A geometric solution to the boundary-avoidance problem in neural networks. Nature, 616(7958):767-775, 2023.